# Multiclass Boosting:
# Simple and Intuitive Weak Learning Criteria

**Nataly Brukhim**
Department of Computer Science
Princeton University

**Amit Daniely**
Department of Computer Science
Hebrew University
Google Research

**Yishay Mansour**
Department of Computer Science
Tel Aviv University
Google Research

**Shay Moran**
Faculty of Mathematics
Faculty of Computer Science
Faculty of Data and Decision Sciences
Technion
Google Research

## Abstract

We study a generalization of boosting to the multiclass setting. We introduce a weak learning condition for multiclass classification that captures the original notion of weak learnability as being "slightly better than random guessing". We give a simple and efficient boosting algorithm, that does not require realizability assumptions and its sample and oracle complexity bounds are independent of the number of classes.

In addition, we utilize our new boosting technique in several theoretical applications within the context of List PAC Learning. First, we establish an equivalence to weak PAC learning. Furthermore, we present a new result on boosting for list learners, as well as provide a novel proof for the characterization of multiclass PAC learning and List PAC learning. Notably, our technique gives rise to a simplified analysis, and also implies an improved error bound for large list sizes, compared to previous results.

## 1 Introduction

Boosting is a powerful algorithmic approach used to boost the accuracy of weak learning models, transforming them into strong learners. Boosting was first studied in the context of binary classification in a line of seminal works which include the celebrated Adaboost algorithm, as well an many other algorithms with various applications (see e.g. [14, 22, 11, 12]).

The fundamental assumption underlying boosting is that a method already exists for finding poor, yet not entirely trivial classifiers. Concretely, binary boosting assumes there exists a learning algorithm that, when presented with training examples, can find a classifier $h : \mathcal{X} \mapsto \{0, 1\}$ that has classification error less than $1/2$. That is, it performs slightly better than random guessing. The intuition is that this is the most minimal assumption one can make about a learning algorithm, without it being impractical. This assumption is called the *weak learning assumption*, and it is central to the study of boosting.

While binary boosting theory has been extensively studied, extending it to the multiclass setting has proven to be challenging. In particular, it turns out that the original notion of weak learnability as being "slightly better than a random guess", does not easily extend to the multiclass case. For

37th Conference on Neural Information Processing Systems (NeurIPS 2023).

example, perhaps the most natural extension is to assume that the learner has accuracy that is slightly better than $1/k$, where $\mathcal{Y} = \{1, ..., k\}$. However, this naive extension is in fact known to be too weak for boosting (see Section 2 below for a detailed discussion). Instead, previous works [19, 7, 23] have formulated various complex weak learning assumptions with respect to carefully tailored loss functions, and rely on restrictive realizability assumptions, making them less useful in practice.

**A weak learning assumption.** In this work, we generalize the classic formulation of boosting to the multiclass setting. We introduce a weak learning condition that captures the original intuition of weak learnability as "slightly better-than-random guessing". The key idea that renders this condition useful compared to previous attempts, is based on a "hint" given to the weak learner. The hint takes the form of a list of $k$ labels per example, where $k$ is possibly smaller than $|\mathcal{Y}|$. Then, the assumption is that there exists a learner capable of producing not entirely trivial classifiers, if it was provided with a "good hint". In other words, if the list provided to the learner happens to contain the correct label, we expect the learner to perform slightly better than randomly guessing a label from the list. Specifically, the assumption is that for any $k \geq 2$, if the given lists of size $k$ contain the true labels, the learner will output a classifier $h : \mathcal{X} \mapsto \mathcal{Y}$ with error slightly better than random guessing among the $k$ labels. Notice that this encompasses both the binary case when $k = 2$, as well as the naive extension mentioned above, when $k = |\mathcal{Y}|$. We call this new condition the "better-than-random guess", or *BRG* condition.

The BRG condition also generalizes the classic binary case condition in a practical sense. Previous methods on multiclass boosting are framed within the PAC (Probably Approximately Correct) setting, correspond to a *known* hypothesis class $\mathcal{H} \subseteq \mathcal{Y}^{\mathcal{X}}$, and assume that weak learning hold for every distribution $\mathcal{D}$ over the *entire* domain $\mathcal{X}$. Practically, these requirements can be very difficult to check or guarantee. In contrast, as in binary boosting, the BRG condition can be relaxed to a more benign empirical weak learning assumption, that can be verified immediately in an actual learning setting.

**Recursive boosting.** Our main contribution is a new boosting algorithm that is founded on the BRG condition. Our boosting methodology yields a simple and efficient algorithm. It is based on the key observation that even a naive weak learner can produce a useful hint. Recall that when given no hint at all, the naive weak learner can still find a hypothesis with a slight edge ($\gamma > 0$), over random guessing among $|\mathcal{Y}|$ labels. Although this may result in a poor predictor, we prove that it effectively reduces the label space per example to approximately $1/\gamma$ labels. This initial hint serves as the starting point for subsequent iterations of the boosting algorithm. The process continues recursively until the label list per example is reduced to size 2, at which point any classic binary boosting can yield a strong classifier. Unlike previous methods, our boosting algorithm and guarantees do not rely on realizability assumptions nor do they scale with $|\mathcal{Y}|$. In fact, we show that the sample and oracle-call complexity of our algorithm are entirely independent of $|\mathcal{Y}|$, which implies our approach is effective even in cases where the label space $\mathcal{Y}$ is possibly infinite. Moreover, the overall running time of our algorithm is polynomial in the size of its input.

An important insight that underlies our approach is the link between the naive weak learning condition, which we term *weak-BRG* learning, to that of *list learning*. In list learning [5, 8, 18], rather than predicting a single outcome for a given unseen input, the goal is to provide a short list of predictions. Here we use this technique as an intermediate goal of the algorithm, by which effectively reducing the size of the label space in each round. The generalization analysis relies on sample compression arguments which result in efficient bounds on the sample and oracle complexities.

Perhaps surprisingly, the connection between weak learnability and list learnability is even more fundamental. We prove that there is an equivalence between these two notions. Specifically, we establish that a $\gamma$-weak learner is equivalent to an $(1/\gamma)$-list learner.

Lastly, we demonstrate the strength of our boosting framework. First, we give a generalization of our boosting technique to hold for list PAC learning algorithms. Then, we showcase the effectiveness of the weak learning criteria in capturing learnability in two fundamental learning settings: PAC learning, and List PAC learning. Recently, [5] proved a characterization of multiclass PAC learning using the Daniely-Shwartz (DS) dimension. In a subsequent study, [8] gave a characterization of *list* learnability using a natural extension of the DS dimension. Here we show that in both cases, assuming the appropriate dimension is bounded, one can devise a simple weak learning algorithm. Thus, it is also amenable to a boosting method similarly to our approach, leading to a novel and alternative proof of the characterization of learnability. We note that for cases where the dimension

is much smaller than the list size, we have an improved result over previous bound. Moreover, our approach offers a simpler algorithm and analysis technique, potentially benefiting future applications as well.

## 1.1 Main result

The main contributions in this work are as follows.

1. **Multiclass boosting framework.** Our main result is a boosting framework for the multiclass setting, which is a natural generalization of binary boosting theory. We give a simple weak learning assumption that retains the notion of weak learnability as "slightly-better-than-random-guess" from the binary case. Furthermore, we give an efficient multiclass boosting algorithm, as formally stated in Theorem 1 below. Our boosting algorithm is given in Section 3 (Algorithm 3).

2. **Applications: List PAC learning.** First, we establish an equivalence between List PAC learning and Weak PAC learning, demonstrating the strong ties between List PAC learning and multiclass boosting theory. Furthermore, we present a new result on boosting for list learners. Lastly, we give a novel and alternative proof for characterization of PAC learning and List PAC learning. In particular, the results imply a simplified algorithmic approach compared to previous works, and improved error bound for cases where the list size is larger than the appropriate dimension [5, 8].

We will now introduce the main weak learning assumption, which we call the "better-than-random guess", or BRG condition, and state our main result in Theorem 1 below.

In its original form, the boosting question begins by assuming that a given hypothesis class $\mathcal{H} \subseteq \{0,1\}^{\mathcal{X}}$ is *weakly-PAC* learnable. Similarly, here we present the BRG condition framed as weak (multiclass) PAC setting, followed by a relaxation to an *empirical* variant of the assumption.

**Definition 1** (BRG condition). *We say that an hypothesis $h : \mathcal{X} \to \mathcal{Y}$ satisfies the $\gamma$-BRG condition with respect a list function $\mu : \mathcal{X} \to \mathcal{Y}^k$ on a distribution $\mathcal{D}$ over examples if*

$$\Pr_{(x,y)\sim\mathcal{D}}[h(x) = y] \geq \left(\frac{1}{k} + \gamma\right) \Pr_{(x,y)\sim\mathcal{D}}[y \in \mu(x)]. \tag{1}$$

*We say that a learning rule $\mathcal{W}$ satisfies the $\gamma$-BRG condition for a hypothesis class $\mathcal{H}$ if for every $\mathcal{H}$-realizable distribution $\mathcal{D}$, for every $k \geq 2$, for every list function $\mu : \mathcal{X} \to \mathcal{Y}^k$, the output hypothesis $h$ outputted by $\mathcal{W}$ satisfies Equation (1) with probability $1 - \delta$, when given $m_0(\delta)$ i.i.d. examples from $\mathcal{D}$, and given $\mu$.*

In words, the condition determines that if $y$ belongs to the set $\mu(x)$, then $h$ has a higher probability of correctly classifying $x$ by an additional factor of $\gamma$, compared to a random guess from the list $\mu(x)$.

However, requiring that the labels be deterministic according to a target function from a known class $\mathcal{H}$, and that weak learning hold for every distribution $\mathcal{D}$ over the entire domain $\mathcal{X}$ are impractical, as they can be very difficult to check or guarantee.

Instead, as in the binary boosting setting, our condition can be relaxed to a more benign *empirical* weak learning assumption, as given next.

**Definition 2** (Empirical BRG condition). *Let $S \in (\mathcal{X} \times \mathcal{Y})^m$. We say that a learning rule $\mathcal{W}$ satisfies the empirical $\gamma$-BRG condition for $S$ if there is an integer $m_0$ such that for every distribution $p$ over $[m]$, for every $k \geq 2$, for every list function[1] $\mu : \mathcal{X}|_S \to \mathcal{Y}^k$, when given $m_0$ examples from $S$ drawn i.i.d. according to $p$, and given $\mu$, it outputs a hypothesis $h$ such that,*

$$\sum_{i=1}^{m} p_i \cdot \mathbb{1}[h(x_i) = y_i] \geq \left(\frac{1}{k} + \gamma\right) \sum_{i=1}^{m} p_i \cdot \mathbb{1}[y_i \in \mu(x_i)]. \tag{2}$$

Next, we give our main result of an efficient boosting algorithm, as stated in Theorem 1 below.

---

[1]We denote $\mathcal{X}|_S = \{x \in \mathcal{X} : \exists y \in \mathcal{Y} \text{ s.t. } (x,y) \in S\}$.

**Theorem 1** (**Boosting** (Informal)). *There exists a multiclass boosting algorithm $\mathcal{B}$ such that for any $\epsilon, \delta > 0$, and any distribution $\mathcal{D}$ over $\mathcal{X} \times \mathcal{Y}$, when given a training set $S \sim \mathcal{D}^m$ and oracle access to a learning rule $\mathcal{W}$ where[2] $m = \tilde{O}\left(\frac{m_0}{\gamma^3 \epsilon}\right)$ and applying $\mathcal{B}$ with a total of $\tilde{O}\left(1/\gamma^3\right)$ oracle calls to $\mathcal{W}$, it outputs a predictor $\bar{H} : \mathcal{X} \mapsto \mathcal{Y}$ such that with probability at least $1 - \delta$, we get that if $\mathcal{W}$ satisfies the empirical $\gamma$-BRG condition for $S$ then,*

$$\Pr_{(x,y)\sim\mathcal{D}}\left[\bar{H}(x) \neq y\right] \leq \epsilon.$$

## 1.2 Related work

Boosting theory has been extensively studied, originally designed for binary classification (e.g., AdaBoost and similar variants) [23]. There are various extension of boosting to the multiclass setting.

The early extensions include AdaBoost.MH, AdaBoost.MR, and approaches based on Error-Correcting Output Codes (ECOC) [24, 1]. These works often reduce the $k$-class task into a single binary task. The binary reduction can have various problems, including increased complexity, and lack of guarantees of an optimal joint predictor.

Other works on multiclass boosting focus on practical considerations and demonstrate empirical performance improvements across various applications [29, 16, 15, 3, 6, 4, 21]. However, they lack a comprehensive theoretical framework for the multiclass boosting problem and often rely on earlier formulations such as one-versus-all reductions to the binary setting or multi-dimensional predictors and codewords.

Notably, a work by [19] established a theoretical framework for multiclass boosting, which generalizes previous learning conditions. However, this requires the assumption that the weak learner minimizes a complicated loss function, that is significantly different from simple classification error. Moreover, it is based on a restrictive realizability assumption with respect to a *known* hypothesis class. In contrast, we do not require realizability, and only consider the standard classification loss.

More recently, [7] followed a formulation for multiclass boosting similar to that of [19]. They proved a hardness result showing that a broad, yet restricted, class of boosting algorithms must incur a cost which scales polynomially with $|\mathcal{Y}|$. Our approach does not fall in this class of algorithms. Moreover, our algorithm has sample and oracle complexity bounds that are entirely independent of $|\mathcal{Y}|$.

## 2 Warmup: *too-weak* weak learning

When there are only 2 labels, the weak learner must find a hypothesis that predicts the correct label a bit better than a random guess. That is, with a success probability that is slightly more than $1/2$. When the number of labels $k$ is more than 2, perhaps the most natural extension requires that the weak learner outputs hypotheses that predict the correct label a bit better than a random guess *among $k$ labels*. That is, with a success probability that is slightly more than $1/k$.

However, this is in fact known to be too weak for boosting (see e.g., [23], Chapter 10). Here we first give a simple example that demonstrates that fact. However, we also show that all is not yet lost for the "better-than-random-guess" intuition. Specifically, we describe how this condition can still allow us to extract valuable knowledge about which labels are *incorrect*. This observation will serve as a foundation for our main results, which we will elaborate on in the next section.

We start by defining the notion of better-than-random weak learner that we term weak-BRG learning.

**Definition 3** (weak-BRG learning). *A learning algorithm $\mathcal{W}$ is a weak-BRG learner for a hypothesis class $\mathcal{H} \subseteq [k]^{\mathcal{X}}$ if there is $\gamma > 0$ and $m_0 : (0, 1) \mapsto \mathbb{N}$ such that for any $\delta_0 > 0$, and any $\mathcal{H}$-realizable distribution $\mathcal{D}$ over $\mathcal{X} \times [k]$, when given $m_0 \geq m_0(\delta_0)$ samples from $\mathcal{D}$, it returns $h : \mathcal{X} \to \mathcal{Y}$ such that with probability $1 - \delta_0$,*

$$\Pr_{(x,y)\sim D}[h(x) = y] \geq \frac{1}{k} + \gamma. \tag{3}$$

To get an intuition for why this definition is indeed too weak for boosting, consider the following simple example. Suppose that $\mathcal{X} = \{a, b, c\}$, $\mathcal{Y} = \{1, 2, 3\}$, and that the training set consists of the

---

[2]The $\tilde{O}$ notation conceals $\mathrm{polylog}(m, 1/\delta)$ factors.

three labeled examples $(a, 1)$, $(b, 2)$, and $(c, 3)$. Further, we suppose that we are using a weak learner which chooses weak classifiers that never distinguish between $a$ and $b$. In particular, the weak learner always chooses one of two weak classifiers: $h_1$ and $h_2$, defined as follows. For $x \in \{a, b\}$ then $h_1$ always returns 1 and $h_2$ always returns 2. For $x = c$ they both return 3.

Then, notice that for any distribution over the training set, either $h_1$ or $h_2$ must achieve an accuracy of at least $1/2$, which is significantly higher than the accuracy of $1/k = 1/3$. However, regardless of how weak classifiers are aggregated, any final classifier $H$ that relies solely on the predictions of the weak hypotheses will unavoidably misclassify either $a$ or $b$. As a result, the training accuracy of $H$ on the three examples can never exceed $2/3$, making it impossible to achieve perfect accuracy through any boosting method.

Furthermore, we note that this simple example can also be extended to a case where the data is realizable by a hypothesis class which is not learnable by any learning algorithm (let alone boosting). For example, consider the hypothesis class $\mathcal{H} = \{1, 2, 3\}^{\mathcal{X}}$ for $\mathcal{X} = \mathbb{N}$. Then, $\mathcal{H}$ is not PAC learnable (e.g., via No-Free-Lunch ([26], Theorem 5.1)). However, similarly as above, one can construct a learning rule that returns a hypothesis with accuracy $1/2 > 1/k$ over an $\mathcal{H}$-realizable distribution.

Next, we will examine a useful observation that will form the basic building block of our algorithmic methodology. We demonstrate that the natural weak learner given in Definition 3, while weak, is nonetheless useful. This can be shown by examining the guarantees obtained through its application in boosting. Specifically, we consider the following classic variant of boosting via the Hedge algorithm.

---

**Algorithm 1** Boosting via Hedge

---

**Given:** Training data $S \in (\mathcal{X} \times [k])^m$, parameter $\eta > 0$.
**Output:** A predictor $H : \mathcal{X} \times \mathcal{Y} \mapsto \mathbb{R}$.

1: Initialize: $w_1(i) = 1$ for all $i = 1, ..., m$.
2: **for** $t = 1, \ldots, T$ **do**
3:    Denote by $\mathcal{D}_t$ the distribution over $[m]$ obtained by normalizing $w_t$.
4:    Draw $m_0$ examples from $\mathcal{D}_t$ and pass to the weak learner.
5:    Get weak hypothesis $h_t : \mathcal{X} \mapsto \mathcal{Y}$, and update for $i = 1, ..., m$:

$$w_{t+1}(i) = w_t(i) e^{-\eta \cdot \mathbb{1}[h_t(x_i) = y_i]}.$$

6: **end for**
7: Output $H$ such that for all $(x, y) \in \mathcal{X} \times [k]$,

$$H(x, y) = \sum_{t=1}^{T} \mathbb{1}[h_t(x) = y].$$

---

Notice that the output of Algorithm 1 returns a predictor that is not a classifier, but a scoring function with the aim of predicting the likelihood of a given label candidate $y \in [k]$ for some $x \in \mathcal{X}$. Typically, boosting algorithms combine the weak hypothesis into such a scoring function yet their final output applies an *argmax* over it, to yield a valid classifier. However, since the weak learning assumption is too weak as we have shown above, taking the argmax is useless in this setting.

Instead, the following lemma shows that by boosting the "too-weak" learner, we can guarantee to eliminate one label for each example in the data. Towards that end, we consider a relaxed variant of the weak-BRG learner, to be defined over a data set $S$, which we term the *empirical weak-BRG* learner. Specifically, we say that a learner satisfies the empirical weak-BRG condition if there is an integer $m_0$ such that for any distribution over the training examples, when given $m_0$ examples drawn i.i.d from it, the learner outputs a hypothesis that satisfies Equation (3).

Proofs are deferred to the appendix.

**Lemma 1** (Remove one label). *Let $S \in (\mathcal{X} \times [k])^m$. Let $\mathcal{W}$ be an empirical weak-BRG learner for $S$ with respect to some $\gamma$ and sample size $m_0$. Then, the output $H : (\mathcal{X} \times [k]) \mapsto [0, T]$ obtained by running Algorithm 1 with $T \geq \frac{8 \log(m)}{\gamma^2}$ and $\eta = \sqrt{\frac{\ln(m)}{2T}}$, guarantees that for all $(x, y) \in S$,*

$\frac{H(x,y)}{T} \geq \frac{1}{k} + \frac{\gamma}{2}$. *Moreover, for all* $(x,y) \in S$ *the minimally scored label* $\hat{\ell} = \arg\min_{\ell \in [k]} H(x, \ell)$ *must be incorrect. That is* $\hat{\ell} \neq y$.

Notice that if we were to take the argmax of $H$ as is typically done in boosting, the guarantees given in Lemma 1 do not suggest this will result in the correct prediction. In fact, this approach might yield a rather bad classifier even for the set $S$ on which it was trained. In other words, for any $(x,y) \in S$ it may be that there is some incorrect label $y' \neq y$ with $H(x, y') > H(x, y)$.

However, notice that the lemma does suggest a good classifier of *incorrect* labels. That is, the lowest scored label will always be an incorrect one, over the training data. This property can be shown to generalize via compression arguments, as discussed in Section 5. This allows us to effectively reduce the size of the label space by one, and is used as the basic building of our algorithm, as detailed in the next section.

## 3  Multiclass boosting results

We start by introducing the notion of weak learnability that is assumed by our boosting algorithm. We note that it is a relaxation of the empirical BRG condition introduced in Definition 2 in the sense that it does not make any guarantees for the case that the given hint list does not contain the correct label. This may seem like significantly weakening the assumption, yet it turns out to be sufficient for our boosting approach to hold.

In the resulting fully relaxed framework, no assumptions at all are made about the data. Although the BRG condition is not explicitly assumed to hold, when this is the case, our final bound given in Theorem 2 implies a high generalization accuracy.

**Definition 4** (Relaxed Empirical $\gamma$-BRG learning). *Let* $S \in (\mathcal{X} \times \mathcal{Y})^m$, $\gamma > 0$, *and integer* $m_0$. *Let* $M$ *be a set of list functions of the form* [3] $\mu : \mathcal{X} \mapsto \mathcal{Y}^k$ *for any integer* $k$, *such that for each* $\mu \in M$ *and* $i \in [m]$, *then* $y_i \in \mu(x_i)$. *A learning algorithm satisfies this condition with respect to* $(S, \gamma, m_0, M)$, *if for any distribution* $p$ *over* $S$ *and any* $\mu : \mathcal{X} \mapsto \mathcal{Y}^k$ *such that* $\mu \in M$, *when given a sample* $S' \sim p^{m_0}$ *and access to* $\mu$, *it returns* $h : \mathcal{X} \mapsto \mathcal{Y}$ *such that,*

$$\Pr_{(x,y) \sim p}[h(x) = y] \geq \frac{1}{k} + \gamma. \tag{4}$$

Notice that when the list $\mu$ returns the set of all possible labels $\mathcal{Y}$ and it is of size $k$, this condition is essentially equivalent to the empirical weak-BRG condition, which as shown above is too weak for boosting. Requiring that the condition will hold for *any* list size $k \leq |\mathcal{Y}|$ is sufficient to facilitate boosting, as shown in Theorem 2.

The starting point of our overall boosting algorithm (given in Algorirhm 3), is a simple learning procedure specified in Algorithm 4 that is used to effectively reduce the size of the label space. In particular, it is used to produce the initial "hint" function that is used by the boosting method.

---

**Algorithm 2** Initial hint

**Given:** $S \in (\mathcal{X} \times \mathcal{Y})^m$, parameters $m_0, p > 0$.
**Output:** A function $\mu : \mathcal{X} \mapsto \mathcal{Y}^p$.

1: Set $S_1 := S$.
2: **for** $j = 1, ..., p$ **do**
3:    Let $\mathcal{U}_j$ denote the uniform distribution over $S_j$.
4:    Draw $m_0$ examples from $\mathcal{U}_j$ and pass to the weak learner, with $\mu_0 \equiv \mathcal{Y}$.
5:    Get weak hypothesis $h_j : \mathcal{X} \mapsto \mathcal{Y}$.
6:    Set $S_{i+1}$ to be all the points in $S_i$ which $h_j$ predicts incorrectly.
7: **end for**
8: Output $\mu$ defined by:
$$\mu(x) = \{h_1(x), \ldots, h_p(x)\}.$$

---

[3] We also allow lists that return an infinite subset of labels. In that case we simply have that the weak hypothesis satisfies $\Pr_{(x,y) \sim p}[h(x) = y] \geq \gamma$.

We can now present our main boosting method in Algorithm 3, and state its guarantees in Theorem 2.

---

**Algorithm 3** Recursive Boosting

---

**Given:** Training data $S \in (\mathcal{X} \times \mathcal{Y})^m$, edge $\gamma > 0$, parameters $T, \eta, p > 0$.
**Output:** A predictor $\bar{H} : \mathcal{X} \mapsto \mathcal{Y}$.

1: Initialize: get $\mu_1$ by applying Algorithm 2 over $S$.
2: **for** $j = 1, \ldots, p - 1$ **do**
3:     Call Hedge (Algorithm 1) with $S$ and $\mu_j$, and parameters $\eta, T$ to get $H_j : \mathcal{X} \times \mathcal{Y} \mapsto \mathbb{R}$.

        \\ Modify Algorithm 1 to receive $\mu_j$ as input,
        and in line 4 pass it to the weak learner.

4:     Construct $\mu_{j+1} : \mathcal{X}|_S \mapsto \mathcal{Y}^{p-j}$ such that for all $x$,

$$\mu_{j+1}(x) = \left\{ y \; : \; y \in \mu_j(x) \; \wedge \; H_j(x,y) > \frac{T}{p - j + 1} \right\},$$

5: **end for**
6: Output the final hypothesis $\bar{H} := \mu_p$.

---

The following theorem is formally stating the main result given in Theorem 1.

**Theorem 2** (Boosting). *Let $\mathcal{W}$ denote a learning rule that when given any set of labeled examples and a list function, returns some hypothesis $h : \mathcal{X} \mapsto \mathcal{Y}$. Let $\epsilon, \delta, \gamma, m_0 > 0$, and let $\mathcal{D}$ a distribution over $\mathcal{X} \times \mathcal{Y}$. Then, when given a sample $S \sim \mathcal{D}^m$ for $m \geq \frac{10^2 \, m_0 \left( \ln^2(m) \ln(\frac{m}{\delta}) \right)}{\gamma^3 \, \epsilon}$, oracle access to $\mathcal{W}$ and $T \geq \frac{8 \ln(m)}{\gamma^2}$, $p \geq \frac{2 \ln(m)}{\gamma}$, and $\eta = \sqrt{\frac{\ln(m)}{2T}}$, Algorithm 3 outputs a predictor $\bar{H}$ such that the following holds. Denote by $M$ the sets of list functions on which $\mathcal{W}$ was trained throughout Algorithm 3. Then, with probability at least $1 - \delta$, we get that if $\mathcal{W}$ satisfies the $\gamma$-BRG condition (as given in Definition 4) with respect to $(S, \gamma, m_0, M)$ then,*

$$\Pr_{(x,y) \sim \mathcal{D}} \left[ \bar{H}(x) \neq y \right] \leq \epsilon.$$

Observe that Theorem 2 implicitly assumes that the sample complexity of the weak learner $m_0$ is not strongly dependent on the overall sample size $m$ and scales at most poly-logarithmically with $m$. In other words, although the statement holds for any $m_0$, the result becomes vacuous otherwise.

In addition, notice that Theorem 2 is quite agnostic in the sense that we have made no prior assumptions about the data distribution. Concretely, Theorem 2 tells us that the generalization error will be small *if* the given oracle learner $\mathcal{W}$ happens to satisfy the $\gamma$-BRG condition with respect to the particular inputs it receives throughout our boosting procedure.

**Adaptive boosting**    Boosting algorithms typically do not assume knowing the value of $\gamma$ and are adapted to it on the fly, as in the well-known Adaboost algorithm [23]. However, the boosting algorithm given in Algorithm 1, as well as our boosting method as a whole, requires feeding the algorithm with a value estimating $\gamma$. If the estimation of $\gamma$ provided to the algorithm is too large, the algorithm may fail. This can be resolved by a simple preliminary binary-search-type procedure, in which we guess gamma, and possible halve it based on the observed outcome. This procedure only increases the overall runtime by a logarithmic factor of $O(\ln(1/\gamma))$, and has no affect on the sample complexity bounds.

## 4   Applications to List PAC learning

The applications given in this section are based on the framework of *List PAC learning* [5, 8], and demonstrate that it is in fact closely related to the multiclass boosting theory. First, we establish an equivalence between list learnability and weak learnability in the context of the PAC model. Furthermore, we present a new result on boosting for list PAC learners. Lastly, we give a novel and alternative proof for characterization of PAC learnability and List PAC learnability. In particular, these imply a simplified algorithmic approach compared to previous works [5, 8].

We start with introducing list learning in Definition 5, followed by the definition of weak PAC learning, similarly to the weak-BRG learning definition we give in this work.

**Definition 5** ($k$-List PAC Learning)**.** *We say that a hypothesis class $\mathcal{H} \subseteq \mathcal{Y}^{\mathcal{X}}$ is $k$-list PAC learnable, if there is an algorithm such that for every $\mathcal{H}$-realizable distribution $\mathcal{D}$, and every $\epsilon, \delta > 0$, when given $S \sim \mathcal{D}^m$ for $m \geq m(\epsilon, \delta)$, it returns $\mu_S : \mathcal{X} \to \mathcal{Y}^k$ such that with probability $1 - \delta$,*

$$\Pr_{(x,y)\sim\mathcal{D}}\left[y \in \mu_S(x)\right] \geq 1 - \epsilon.$$

**Definition 6** ($\gamma$-weak PAC Learning)**.** *We say that a hypothesis class $\mathcal{H} \subseteq \mathcal{Y}^{\mathcal{X}}$ is $\gamma$-weak PAC learnable, if there is an algorithm such that for every $\mathcal{H}$-realizable distribution $\mathcal{D}$, and every $\delta > 0$, when given $S \sim \mathcal{D}^m$ for $m \geq m(\delta)$, it returns $h_S : \mathcal{X} \to \mathcal{Y}$ such that with probability $1 - \delta$,*

$$\Pr_{(x,y)\sim\mathcal{D}}\left[y = h_S(x)\right] \geq \gamma.$$

Next, in the following lemmas we show the strong connection between these two notions. Specifically, we give an explicit construction of a list learner given oracle access to a weak learner, and vice versa.

**Lemma 2** (Weak $\Rightarrow$ List Learning)**.** *Let $\mathcal{H} \subseteq \mathcal{Y}^{\mathcal{X}}$ be a hypothesis class. Assume $\mathcal{W}$ is a $\gamma$-weak PAC learner for $\mathcal{H}$ with sample complexity $m_w : (0, 1) \mapsto \mathbb{N}$. Let $k$ be the smallest integer such that $\frac{1}{k} < \gamma$, and denote $\sigma = \gamma - \frac{1}{k}$. Then, there is an $(k-1)$-List PAC learner with sample complexity $m(\epsilon, \delta) = \tilde{O}\left(\frac{m_w(\delta/T)}{\sigma^2\epsilon}\right)$ where $T = \tilde{O}(\frac{1}{\sigma^2})$ is the number of its oracle calls to $\mathcal{W}$.*

**Lemma 3** (List $\Rightarrow$ Weak Learning)**.** *Let $\mathcal{H} \subseteq \mathcal{Y}^{\mathcal{X}}$ be a hypothesis class. Assume $\mathcal{L}$ is a $k$-List PAC learner for $\mathcal{H}$ with sample complexity $m_\ell : (0, 1) \mapsto \mathbb{N}$. Then, for any $\epsilon > 0$ there is an $\gamma$-Weak PAC learner where $\gamma = \frac{1-2\epsilon}{k}$, with sample complexity $m(\delta) = \tilde{O}\left(m_\ell(\epsilon, 1/2) \cdot k + (k/\epsilon)^2\right)$ where $q = 2k\log(2/\delta)$ is the number of its oracle calls to $\mathcal{L}$.*

Lastly, Theorem 3 concludes this section demonstrating the strong ties between weak and list learnability. Concretely, it combines the results of both Lemma 2 and Lemma 3 above to show that when the appropriate parameters $\gamma$ and $k$ are optimal, then $\gamma$-PAC learnability and $k$-list PAC learnability are in fact equivalent.

**Theorem 3** (Optimal accuracy $\iff$ Optimal list size)**.** *Let $\mathcal{H} \subseteq \mathcal{Y}^{\mathcal{X}}$. Denote by $k(\mathcal{H})$ the smallest integer $k$ for which $\mathcal{H}$ is $k$-list PAC learnable, assuming that $k(\mathcal{H}) < \infty$. Denote by $\gamma(\mathcal{H})$ the supremum over $\gamma \in [0, 1]$ for which $\mathcal{H}$ is $\gamma$-weak PAC learnable. Then, it holds that $k(\mathcal{H}) \cdot \gamma(\mathcal{H}) = 1$.*

## 4.1 List boosting and conformal learning

List prediction rules naturally arise in the setting of *conformal learning*. In this model, algorithms make their predictions while also offering some indication of the level of reliable confidence in those predictions. For example in multiclass classification, given an unlabeled test point $x$, the conformal learner might output a list of all possible classes along with scores which reflect the probability that $x$ belongs to each class. This list can then be truncated to a shorter one which contains only the classes with the highest score. See the book by [28] and surveys by [25, 2] for more details.

We now consider a closely related notion of List PAC learnability, that similarly to conformal learning allows the list size to depend on the desired confidence. This was also defined in [8], termed underline{weak} List PAC Learning, due to the dependence of the list size on the input parameter.

Indeed, it is natural to expect that the list size will increase when we require a more refined accuracy, and perhaps that this is a weaker notion of learnability than that of List PAC learning, which corresponds to a fixed list size.

Interestingly, it turns out that weak List PAC Learning is in fact equivalent to strong List PAC Learning. In other words, a list learner with a list size that varies with the desired accuracy parameter can be *boosted* to a list learner with a fixed list size, and arbitrarily good accuracy. The proof is by way of a generalization of our boosting technique to lists, as stated in Theorem 4.

**Theorem 4** (List boosting)**.** *Let $\mathcal{H} \subseteq \mathcal{Y}^{\mathcal{X}}$. Let $\epsilon_0, \delta_0 > 0$, and assume that there exists an algorithm such that for every $\mathcal{H}$-realizable distribution $\mathcal{D}$, and for some integer $k_0 := k_0(\epsilon_0)$, when given $S \sim \mathcal{D}^m$ for $m \geq m(\epsilon_0, \delta_0)$, it returns $\mu_S : \mathcal{X} \to \mathcal{Y}^{k_0}$ such that with probability $1 - \delta_0$,*

$$\Pr_{(x,y)\sim\mathcal{D}}\left[y \in \mu_S(x)\right] \geq 1 - \epsilon_0.$$

*Then, there is a k-List PAC learning algorithm for $\mathcal{H}$ for a* fixed *list size* $k = \left\lfloor \frac{k_0}{1 - 2\epsilon_0} \right\rfloor$.

Observe that Theorem 4 indeed generalizes classic boosting. Specifically, consider the binary setting and notice that when $k_0 = 1$, and $\epsilon_0$ is slightly smaller than $1/2$, Theorem 4 implies that weak learning with edge $\approx \frac{1}{2} - \epsilon_0$, is equivalent to strong learning with arbitrarily small error. The following corollary shows that weak List PAC Learning implies strong List PAC Learning.

**Corollary 1.** *If a class $\mathcal{H} \subseteq \mathcal{Y}^{\mathcal{X}}$ is weakly-List PAC learnable then it is also List PAC learnable.*

### 4.2 Characterization of List PAC learnability

We now focus on the characterization of List PAC learnability, which also implies the characterization of PAC learnability. Towards that end, we define the Daniely-Shwartz (DS) dimension [9]. Specifically, we give the natural generalization of it to $k$-sized lists, called the $k$-DS dimension.

**Definition 7** ($k$-DS dimension [8]). *Let $\mathcal{H} \subseteq \mathcal{Y}^{\mathcal{X}}$ be a hypothesis class and let $S \in \mathcal{X}^d$ be a sequence. We say that $\mathcal{H}$ $k$-DS shatters $S$ if there exists $\mathcal{F} \subseteq \mathcal{H}, |\mathcal{F}| < \infty$ such that $\forall f \in \mathcal{F}|_S$, $\forall i \in [d]$, $f$ has at least $k$ $i$-neighbors. The $k$-DS dimension of $\mathcal{H}$, denoted as $d_{DS}^k = d_{DS}^k(\mathcal{H})$, is the largest integer $d$ such that $\mathcal{H}$ $k$-DS shatters some sequence $S \in \mathcal{X}^d$.*

We note that when $k = 1$, this captures the standard DS dimension [9]. We show that when the $k$-DS dimension is bounded, one can construct a simple weak learner which satisfies our BRG condition. Thus, it is also amenable to our boosting method, leading to a qualitatively similar results for characterization of learnability as in [5, 8]. The result is given in the next theorem.

**Theorem 5** (PAC and List-PAC learnability). *Let $\mathcal{H} \subseteq \mathcal{Y}^{\mathcal{X}}$ be an hypothesis class with $k$-DS dimension $d < \infty$. Then, $\mathcal{H}$ is List PAC learnable. Furthermore, there is a learning algorithm $A$ for $\mathcal{H}$ with the following guarantees. For every $\mathcal{H}$-realizable distribution $\mathcal{D}$, every $\delta > 0$ and every integer $m$, given an input sample $S \sim \mathcal{D}^m$, the algorithm $A$ outputs $\mu = A(S)$ such that[4]*

$$\Pr_{(x,y) \sim \mathcal{D}}[\mu(x) \not\ni y] \leq \tilde{O}\left( \frac{d^5 k^4 + \log(1/\delta)}{m} \right),$$

*with probability at least $1 - \delta$ over $S$. In particular, if $k = 1$, then $\mathcal{H}$ is PAC learnable.*

We remark that for cases where $d \ll k$ we have an improved result over the bound given by [8]. For comparison, the error bound given by [8], Theorem 2 is $\tilde{O}\left( \frac{d^{1.5} k^6 + \log(1/\delta)}{m} \right)$.

Thus, Theorem 5 demonstrates that our boosting-based approach gives rise to an alternative proof for the characterization of PAC learnability and List PAC learnability. Moreover, our approach offers a simpler algorithm and analysis technique, that can perhaps be of use in future applications as well.

## 5 Generalization via compression

This section is concerned with the analysis of our boosting method given in Algorithm 3, and the proof of our main result given in Theorem 1 (and formally in Theorem 2).

The boosting algorithm given in this work is best thought of as a *sample compression scheme* [17]. A sample compression scheme (Definition 8) is an abstraction of a common property to many learning algorithms. It can be viewed as a two-party protocol between a *compresser* and a *reconstructor*. The compresser gets as input a sample $S$. The compresser picks a small subsample $S'$ of $S$ and sends it to the reconstructor. The reconstructor outputs an hypothesis $h$. The correctness criteria is that $h$ needs to correctly classify *all* examples in the input sample $S$. We formally define it next.

**Definition 8** (Sample Compression Scheme [17]). *Let $r \leq m$ be integers. An $m \to r$ sample compression scheme* consists of a reconstruction *function*

$$\rho : (\mathcal{X} \times \mathcal{Y})^r \to \mathcal{Y}^{\mathcal{X}}$$

*such that for every $S \in (\mathcal{X} \times \mathcal{Y})^m$, there exists $S' \subseteq S$ of size $r$, such that for all $(x, y) \in S$ it holds that $h(x) = y$, where $h = \rho(S')$.*

---

[4]The $\tilde{O}$ notation conceals $\text{polylog}(m, 1/\gamma)$ factors.

We are now ready to prove the main result, given in Theorem 2. The next paragraph highlights the assumptions that were made, followed by the proof of the theorem.

Specifically, we assume for simplicity that the learning algorithm does not employ internal randomization. Thus, it can be regarded as a fixed, deterministic mapping from a sequence of $m_0$ unweighted examples, and a function $\mu : \mathcal{X} \mapsto \mathcal{Y}^k$, to a hypothesis $h : \mathcal{X} \mapsto \mathcal{Y}$. We note that our results remain valid for a randomized learner as well, yet we assume the above for ease of exposition.

*Proof of Theorem 2.* The proof is given via a sample compression scheme, demonstrating that if weak learnability holds, then the final predictor $\bar{H}$ can be represented using a small number of training examples, and that it is consistent with the entire training set.

First, we fix the sample $S$ and assume that the $\gamma$-BRG condition holds for $S$ as in the theorem statement. We will then show for each $j = 1...p$ that $\mu_{j+1}$ satisfies the following 3 properties: (a) for each $x \in \mathcal{X}$ it returns at most $p - j$ labels, (b) for all $(x, y) \in S$, it holds that $y \in \mu_{j+1}(x)$, and (c) it can be represented using only a small number of training examples.

First, note that $\mu_1$ is indeed a mapping to at most $p$ labels by its construction in Algorithm 2. Moreover, recall that by the above assumption, the weak learner is a deterministic mapping from its input to a hypothesis. Therefore, any hypothesis produced by the weak learner within Algorithm 2 can be represented simply by the sequence of $m_0$ examples on which it was trained. Lemma 4 implies that there is a subset $S' \subsetneq S$ of size at most $m_0 \cdot p$, where $p = \lceil \log(m)/\gamma \rceil$ such that the following holds: There are $p$ hypotheses $h_i' : \mathcal{X} \mapsto \mathcal{Y}$, that comprise the list $\mu_1$, where each $h_i'$ can be represented by $m_0$ examples in $S'$. It is also guaranteed by Lemma 4 that for all $(x, y) \in S$, it holds that $y \in \mu_1(x)$.

Next, we will show that the 3 properties (a)-(c) above holds for $\mu_2$ (and similarly for all $j \geq 2$). Consider the first $T$ weak hypotheses $h_1^{(1)}, ..., h_T^{(1)}$ generated by Algorithm 3, within its first call to Algorithm 1. Notice that each $h_t^{(1)}$ can now be represented by the sequence of $m_0$ examples on which it was trained, as well as the same $m_0 \cdot p$ examples from above that correspond to $\mu_1$. Therefore, we can represent the mapping $\mu_2$ by a total of $T \cdot m_0 + m_0 \cdot p$ examples. Next, we will show that (a) and (b) hold, by applying Lemma 1. Specifically, we use it to prove that for each $(x, y) \in S$, $\mu_2(x)$ returns $p - 1$ labels, and also that $y \in \mu_2(x)$.

We first show that the conditions of Lemma 1 are met, by considering a simple conversion of all the labels according to $\mu_1$. Specifically, since both Lemma 1 and Algorithm 1 assume the labels are in $[k]$, yet both Algorithm 3 and our weak learner assume the labels in $S$ are in $\mathcal{Y}$, we can think of mapping each $y \in \mathcal{Y}$ to $[p + 1]$ according to $\mu_1$, getting its corresponding label $\ell \in [p + 1]$ in the mapped space, and then remapping back to the $\mathcal{Y}$ space when returning to Algorithm 3.

Concretely, for each pair $(x, y) \in S$, convert it to $(x, \ell) \in \mathcal{X} \times [p]$, such that the $\ell$-th entry of $\mu_1(x)$ is $y$, denoted $\mu_1(x)_\ell = y$. By Definition 4 we obtain a hypothesis $h : \mathcal{X} \mapsto \mathcal{Y}$. For its internal use in Algorithm 1, we convert it into a hypothesis $h' : \mathcal{X} \mapsto [p + 1]$ such that if $h(x) \in \mu(x)$, set $h'(x) = \ell$ for $\ell$ that satisfies $\mu_1(x)_\ell = h(x)$, or $h'(x) = p + 1$ if there is no such $\ell \in [p]$. Finally, we set the output $H_1$ of Algorithm 1 to be defined with respect to the original, remapped, weak hypotheses $h$.

Now applying Lemma 1 with $k := p$, we get that for all $(x, y) \in S$, we have $H_1(x, y) > T/p$. Therefore, it must hold that $y \in \mu_2(x)$. Moreover, since $\sum_{y' \in \mu_1(x)} H_1(x, y') \leq \sum_{y' \in \mathcal{Y}} H_1(x, y') \leq T$,

Lemma 1 implies that there must be a label $y' \neq y$ such that $y' \in \mu_1(x)$ for which $H_1(x, y') < T/p$. Therefore, by construction of $\mu_2$ we get that $y' \notin \mu_2(x)$, and $|\mu_2(x)| \leq |\mu_1(x) \setminus \{y'\}| = p - 1$.

Next, we continue in a similar fashion for all rounds $j = 3, ..., p - 1$. Namely, the same arguments as above show that by applying Lemma 1 with $k := p - j + 2$, we get that $\mu_{j+1}$ satisfies the above conditions over $S$. Moreover, to represent each weak hypotheses $h_t^{(j)}$ generated by Algorithm 3 within its $j$-th call to Algorithm 1, we use the sequence of $m_0$ examples on which it was trained, as well as the same $(j - 1) \cdot T \cdot m_0 + m_0 \cdot p$ examples from above that correspond to $\mu_j$.

Overall, we have shown that if $\mathcal{W}$ satisfies the $\gamma$-BRG condition (as given in Definition 4) with respect to $(S, \gamma, m_0, M)$ then the final predictor $\bar{H} := \mu_p$ is both consistent with the sample $S$, and can be represented using only $r$ examples, where,

$$r = (p - 1) \cdot T \cdot m_0 + m_0 \cdot p = O\left(\frac{p \cdot m_0 \ln(m)}{\gamma^2}\right) = O\left(\frac{m_0 \ln^2(m)}{\gamma^3}\right). \tag{5}$$

We can now apply a sample compression scheme bound to obtain the final result. Specifically, we apply Theorem 6 (for $k = 1$), for a $m \to r$ sample compression scheme algorithm $\mathcal{A}$ equipped with a reconstruction function $\rho$ (see Definition 8). We denote $\mathrm{err}_{\mathcal{D}}(\bar{H}) = \mathrm{Pr}_{(x,y) \sim \mathcal{D}}[\bar{H}(x) \neq y]$. Then, by Theorem 6 we get that,

$$\Pr_{S \sim \mathcal{D}^m, \mathcal{A}} \left[ \bar{H} \text{ consistent with } S \Rightarrow \mathrm{err}_{\mathcal{D}}(\bar{H}) > \frac{r \ln(m) + \ln(1/\delta)}{m - r} \right] \leq \delta,$$

where the overall randomness of our algorithm is denoted by $\mathcal{A}$. Plugging in $r$ from Equation (5), and $m$ given in the theorem statement, yields the desired bound. □

## Acknowledgments and Disclosure of Funding

AD received funding from the European Research Council (ERC) under the European Union's Horizon 2022 research and innovation program (grant agreement No. 101041711), and the Israel Science Foundation (grant number 2258/19). This research was done as part of the NSF-Simons Sponsored Collaboration on the Theoretical Foundations of Deep Learning.

YM received funding from the European Research Council (ERC) under the European Union's Horizon 2020 research and innovation program (grant agreement No. 882396), by the Israel Science Foundation (grant number 993/17), the Yandex Initiative for Machine Learning at Tel Aviv University and a grant from the Tel Aviv University Center for AI and Data Science (TAD).

SM is a Robert J. Shillman Fellow; he acknowledges support by ISF grant 1225/20, by BSF grant 2018385, by an Azrieli Faculty Fellowship, by Israel PBC-VATAT, by the Technion Center for Machine Learning and Intelligent Systems (MLIS), and by the the European Union (ERC, GENERALIZATION, 101039692). Views and opinions expressed are however those of the author(s) only and do not necessarily reflect those of the European Union or the European Research Council Executive Agency. Neither the European Union nor the granting authority can be held responsible for them.

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

## A Missing proofs of Section 2

*Proof of Lemma 1.* First, standard analysis of the Hedge algorithm [12] implies that,

$$\sum_{t=1}^{T} \Pr_{i \sim \mathcal{D}_t} [h_t(x_i) = y_i]) \leq \frac{\ln(m)}{\eta} + \eta T + H(x_i, y_i). \tag{6}$$

For completeness we re-prove it here. Denote $\phi_t = \sum_{i=1}^{m} w_t(i)$ and $\alpha_t = \Pr_{i \sim \mathcal{D}_t}[h_t(x_i) = y_i])$, and observe that,

$$\phi_{t+1} = \sum_{i=1}^{m} w_{t+1}(i) = \sum_{i=1}^{m} w_t(i) e^{-\eta \mathbb{1}[h_t(x_i) = y_i]} = \phi_t \sum_{i=1}^{m} \mathcal{D}_t(i) e^{-\eta \mathbb{1}[h_t(x_i) = y_i]}$$

$$\leq \phi_t \sum_{i=1}^{m} \mathcal{D}_t(i)(1 - \eta \mathbb{1}[h_t(x_i) = y_i] + \eta^2) = \phi_t(1 - \eta \alpha_t + \eta^2) \leq \phi_t e^{-\eta \alpha_t + \eta^2},$$

where the two inequalities follows from $e^{-x} \leq 1 - x + x^2$ for $x \geq 0$, and $1 + x \leq e^x$, respectively.

Therefore, after $T$ rounds we get $\phi_T \leq m e^{-\eta \sum_{t=1}^{T} \alpha_t + T\eta^2}$. Furthermore, for every $i \in [m]$ we have,

$$e^{-\eta H(x_i, y_i)} = w_T(i) \leq \phi_T \leq m e^{-\eta \sum_{t=1}^{T} \alpha_t + T\eta^2}.$$

Taking logarithms and re-arranging we get Equation (6), as needed.

Next, we lower bound the left-hand side with $\frac{1}{k} + \gamma$ by the weak learning guarantee. By also substituting $\eta = \sqrt{\frac{\ln(m)}{2T}}$, we get,

$$\frac{1}{k} + \gamma - \sqrt{\frac{2\ln(m)}{T}} \leq \frac{H(x_i, y_i)}{T}.$$

Plugging in $T$ yields the desired bound.

To prove the second part of the lemma, observe that by its first statement it holds that for each $(x, y) \in S$ there must be a label $\ell \neq y$ for which $H(x, \ell) < T/k$. □

The following lemma proves that Algorithm 2 constructs a good initial hint to the boosting algorithm (Algorithm 3).

**Lemma 4** (Initial hint). *Let $m_0, \gamma > 0$, and let $S \in (\mathcal{X} \times \mathcal{Y})^m$. Given oracle access to an empirical $\gamma$-BRG learner for $S$ (Definition 4), when run with $p = \lceil \ln(m)/\gamma \rceil$, Algorithm 2 outputs $\mu$ of size $p$, such that for all $(x, y) \in S$ it holds that $y \in \mu(x)$.*

*Proof.* Observe that for all $j \in [p]$ the output of the weak learner satisfies,

$$\Pr_{(x,y) \sim \mathcal{U}_j} [h_j(x) = y] \geq \gamma, \tag{7}$$

where this follows from Definition 4 . This implies that $|S_p| \leq (1 - \gamma)^p m < e^{-\gamma p} m < 1$. □

## B Missing proofs of Section 4

We say that a list function $\mu : \mathcal{X} \to \mathcal{Y}^k$ is *consistent* with a sample $S \in (\mathcal{X} \times \mathcal{Y})^m$ if for every $(x, y) \in S$, it holds that $y \in \mu(x)$.

We now define a compression scheme for lists as follows. Specifically, we say that an algorithm $\mathcal{A}$ is based on an $m \to r$ compression scheme for $k$-lists if there exists a reconstruction function $\rho : (\mathcal{X} \times \mathcal{Y})^r \to (\mathcal{Y}^k)^{\mathcal{X}}$, such that when $\mathcal{A}$ is provided with a set $S \in (\mathcal{X} \times \mathcal{Y})^m$ of training examples, it somehow chooses (possibly in a randomized fashion) a subset $S' \subsetneq S$ of size $r$, and outputs a $k$-list function $\mu = \rho(S')$. Denote its error with respect to a distribution $\mathcal{D}$ over $\mathcal{X} \times \mathcal{Y}$ as,

$$\text{err}_{\mathcal{D}}(\mu) = \Pr_{(x,y) \sim \mathcal{D}} \left[ y \notin \mu(x) \right].$$

Next, we prove a generalization bound for a compression scheme for $k$-lists. The proof follows similarly to [17, 10], extending these classical generalization bounds to the notion of lists.

**Theorem 6.** *Let $r \leq m$ be integers and $k < |\mathcal{Y}|$. Let $\mathcal{A}$ denote a (possibly randomized) algorithm based on an $m \to r$ compression scheme for $k$-lists. For any $S \in (\mathcal{X} \times \mathcal{Y})^m$ denote by $f_S$ the output of $\mathcal{A}$ over $S$. Let $\delta > 0$ and set $\epsilon = \frac{r \ln(m) + \ln(1/\delta)}{m-r}$. Then, for any distribution $\mathcal{D}$ over $\mathcal{X} \times \mathcal{Y}$, with probability at least $1 - \delta$ it holds that if $f_S$ is consistent with $S$ then, $\mathrm{err}_{\mathcal{D}}(f_S) \leq \epsilon$.*

*Proof.* We want to show that,

$$\Pr_{S \sim \mathcal{D}^m, \mathcal{A}} \left[ f_S \text{ is consistent with } S \Rightarrow \mathrm{err}_{\mathcal{D}}(f_S) \leq \epsilon \right] \geq 1 - \delta. \tag{8}$$

In other words, we want to show that,

$$\Pr_{S \sim \mathcal{D}^m, \mathcal{A}} \left[ f_S \text{ is consistent with } S \wedge \mathrm{err}_{\mathcal{D}}(f_S) > \epsilon \right] \leq \delta. \tag{9}$$

The equivalence of Equations (8) and (9) is because if $A$ and $B$ are events, then $A \Rightarrow B$ is exactly equivalent to $(\neg A) \vee B$ whose negation is $A \wedge (\neg B)$.

We overload notation and for any $S' \in (\mathcal{X} \times \mathcal{Y})^r$ we denote $f_{S'} := \rho(S')$, where $\rho$ is the reconstruction function associated with $\mathcal{A}$.

First, we fix indices $i_1, ..., i_r \in [m]$. Given the random training set $S$. Observe that for *any* list function $f : \mathcal{X} \to \mathcal{Y}^k$ that is $\epsilon$-bad, then the probability of also having $f$ be consistent with $S \setminus S'$ is at most $(1 - \epsilon)^{m-r}$. This holds for any such $f$, and in particular $f_{S'}$ if it happens to be $\epsilon$-bad. Since this also holds for any other fixed choice of indices $i_1, ..., i_r \in [m]$, we can bound the above for all possible sequences of indices. Therefore, by the union bound, since there are $m^r$ choices for these indices, we have that for any $f_S$ chosen by $\mathcal{A}$ it holds that,

$$\Pr_{S \sim \mathcal{D}^m, \mathcal{A}} \left[ f_S \text{ consistent with } S \wedge f_S \text{ is } \epsilon\text{-bad} \right] \tag{10}$$

$$\leq \Pr_{S \sim \mathcal{D}^m} \left[ \exists S' \in S^r : f_{S'} \text{ consistent with } S \setminus S' \wedge f_{S'} \text{ is } \epsilon\text{-bad} \right] \tag{11}$$

$$\leq \sum_{i_1 ... i_r \in [m]} \Pr_{S \sim \mathcal{D}^m} \left[ f_{S'} \text{ consistent with } S \setminus S' \wedge f_{S'} \text{ is } \epsilon\text{-bad} \right] \tag{12}$$

$$= m^r \cdot \Pr_{\substack{S'' \sim \mathcal{D}^{m-r} \\ S' \sim \mathcal{D}^r}} \left[ f_{S'} \text{ consistent with } S'' \wedge f_{S'} \text{ is } \epsilon\text{-bad} \right] \tag{13}$$

$$= m^r \cdot \Pr_{\substack{S'' \sim \mathcal{D}^{m-r} \\ S' \sim \mathcal{D}^r}} \left[ f_{S'} \text{ consistent with } S'' \mid f_{S'} \text{ is } \epsilon\text{-bad} \right] \cdot \Pr_{S' \sim \mathcal{D}^r} \left[ f_{S'} \text{ is } \epsilon\text{-bad} \right] \tag{14}$$

$$\leq m^r \cdot \Pr_{\substack{S'' \sim \mathcal{D}^{m-r} \\ S' \sim \mathcal{D}^r}} \left[ f_{S'} \text{ consistent with } S'' \mid f_{S'} \text{ is } \epsilon\text{-bad} \right] \tag{15}$$

$$= m^r \cdot \mathbb{E}_{S' \sim \mathcal{D}^r} \left[ \Pr_{S'' \sim \mathcal{D}^{m-r}} \left[ f_{S'} \text{ is consistent with } S'' \right] \mid f_{S'} \text{ is } \epsilon\text{-bad} \right] \tag{16}$$

$$\leq m^r (1 - \epsilon)^{m-r}, \tag{17}$$

where in Equation (12) we use the notation $S' = \{(x_{i_1}, y_{i_1}), ..., (x_{i_r}, y_{i_r})\}$, and where the last inequality holds since for any particular selection of the examples $S' \in (\mathcal{X} \times \mathcal{Y})^r$ for which $f_{S'}$ is $\epsilon$-bad, we have that $\Pr_{S'' \sim \mathcal{D}^{m-r}} \left[ f_{S'} \text{ is consistent with } S'' \right] \leq (1 - \epsilon)^{m-r}$. This means that it also holds true if these examples are selected at random. Then, we get the desired bound since $m^r (1 - \epsilon)^{m-r} \leq m^r e^{-\epsilon(m-r)} = \delta$, which follows by solving for $\delta$ via our definition of $\epsilon$ above.

$\square$

**Theorem 7.** *Let $r \leq m$ be integers and $k < |\mathcal{Y}|$. Let $\mathcal{A}$ denote a (possibly randomized) algorithm based on an $m \to r$ compression scheme for $k$-lists. For any $S \in (\mathcal{X} \times \mathcal{Y})^m$ denote by $f_S$ the output of $\mathcal{A}$ over $S$. Let $\delta > 0$ and set $\epsilon = \frac{r \ln(m) + \ln(1/\delta)}{m-r}$. Then, for any distribution $\mathcal{D}$ over $\mathcal{X} \times \mathcal{Y}$,*

$$\Pr_{S \sim \mathcal{D}^m, \mathcal{A}} \left[ \mathrm{err}_{\mathcal{D}}(f_S) > \epsilon \right] \leq \Pr_{S \sim \mathcal{D}^m, \mathcal{A}} \left[ f_S \text{ is not consistent with } S \right] + \delta. \tag{18}$$

*Proof.* Let $C_{\mathcal{A}}(S) = \mathbf{1}\big[\ f_S \text{ is consistent with } S\big]$. First, using the simple fact that for any two events $A$ and $B$ it holds that $\Pr[A] = \Pr[A \wedge B] + \Pr[A \wedge \neg B] \le \Pr[A \wedge B] + \Pr[\neg B]$, we get,

$$\Pr_{S \sim \mathcal{D}^m, \mathcal{A}}\left[f_S \text{ is } \epsilon\text{-bad}\right] \le \Pr_{S \sim \mathcal{D}^m, \mathcal{A}}\left[C_{\mathcal{A}}(S) \ \wedge \ f_S \text{ is } \epsilon\text{-bad}\right] + \Pr_{S \sim \mathcal{D}^m, \mathcal{A}}\left[\neg C_{\mathcal{A}}(S)\right].$$

Thus, it remains to prove that,

$$\Pr_{S \sim \mathcal{D}^m, \mathcal{A}}\left[C_{\mathcal{A}}(S) \ \wedge \ f_S \text{ is } \epsilon\text{-bad}\right] \le \delta, \tag{19}$$

which holds true by Theorem 6 above. $\qquad \square$

---

**Algorithm 4** Boosting Weak-to-List Learning
---
**Given:** training data $S \in (\mathcal{X} \times \mathcal{Y})^m$, edge $\gamma > 0$, parameters $T, \eta > 0$.
**Output:** A list $\mu : \mathcal{X} \mapsto \mathcal{Y}^{k-1}$.

1: Let $k$ be the smallest integer such that $\frac{1}{k} < \gamma$.
2: Call Hedge (Algorithm 1) with $S$ and parameters $\eta, T$ to get $H : \mathcal{X} \times \mathcal{Y} \mapsto [0, T]$.
3: Construct $\mu : \mathcal{X}|_S \mapsto \mathcal{Y}^{k-1}$ such that for all $x$,

$$\mu(x) = \left\{ \ell \in \mathcal{Y} \ : \ H(x, \ell) > \frac{T}{k} \right\}.$$

4: Output $\mu : \mathcal{X} \to \mathcal{Y}^{k-1}$.  \\\\ For $x \notin S$, truncate $\mu(x)$ to first $k-1$ labels, if contains more.

---

The following lemma describes the list-learning guarantees that can be obtained when given access to a $\gamma$-weak learner (Definition 6). It is a re-phrasing of Lemma 2 from the main paper, using exact constants. The next paragraph highlights the assumptions that were made, followed by the proof of the lemma.

Specifically, we assume for simplicity that the learning algorithm does not employ internal randomization. Thus, it can be regarded as a fixed, deterministic mapping from a sequence of $m_0$ unweighted examples, to a hypothesis $h : \mathcal{X} \to \mathcal{Y}$. We note that our results remain valid for a randomized learner as well, yet we assume the above for ease of exposition.

**Lemma 5** (Weak-to-List Boosting). *Let $\mathcal{H} \subseteq \mathcal{Y}^{\mathcal{X}}$ be a hypothesis class. Let $\gamma, \epsilon, \delta > 0$, let $k$ be the smallest integer such that $\frac{1}{k} < \gamma$, and denote $\sigma = \gamma - \frac{1}{k}$. Let $\mathcal{D}$ be an $\mathcal{H}$-realizable distribution over $\mathcal{X} \times \mathcal{Y}$. Then, when given a sample $S \sim \mathcal{D}^m$ for $m \ge \frac{16 \, m_0 \ln^2(m)}{\sigma^2 \epsilon} + \frac{\ln(2/\delta)}{\epsilon}$, oracle access to a $\gamma$-weak learner (Definition 6) for $\mathcal{H}$ with $m_0 \ge m_0(\frac{\delta}{2T})$ and $T \ge \frac{8 \ln(m)}{\sigma^2}$, and $\eta = \sqrt{\frac{\ln(m)}{2T}}$, Algorithm 4 outputs a list function $\mu : \mathcal{X} \to \mathcal{Y}^{k-1}$ such that with probability at least $1 - \delta$,*

$$\Pr_{(x,y) \sim \mathcal{D}}\left[y \notin \mu(x)\right] \le \epsilon.$$

*Proof.* By the learning guarantee in Definition 6, we know that for every distribution $D$ over the examples in $S$, when given $m_0$ examples from $D$, with probability at least $1 - \delta/(2T)$ the learner returns $h : \mathcal{X} \to \mathcal{Y}$ such that

$$\sum_{i=1}^{m} D(i) \mathbb{1}[h(x_i) = y_i] \ge \gamma = \frac{1}{k} + \sigma.$$

In line 2 of Algorithm 4, we call Algorithm 1 with $S$ and the weak learner, with $T$, $\eta$ and $m_0$ as defined above. Then, we follow the same proof as in Lemma 1, applied to $S$ such that the guarantees of the weak learner holds with probability at least $1 - \delta/(2T)$ as above. Then, taking the union bound over all $T$ calls to the weak learner, by the same argument of Lemma 1 we get that with probability at least $1 - \delta/2$ it holds that for all $(x, y) \in S$,

$$\frac{1}{T} \sum_{t=1}^{T} \mathbb{1}[h_t(x) = y] \ge \frac{1}{k} + \frac{\sigma}{2}, \tag{20}$$

where $h_1, ..., h_T$ are the weak hypotheses obtained via Algorithm 1 and that comprise of $H$, the output of Algorithm 1. Moreover, notice that since the weak learning algorithm can be regarded as a fixed, deterministic mapping from a sequence of $m_0$ unweighted examples to a hypothesis, hypothesis $h_t$ can be expressed via these very $m_0$ examples. Notice that $H$ can be represented by $T \cdot m_0$ examples. Thus, the list function $\mu$ defined in Line 4 of Algorithm 4, can also be represented by $T \cdot m_0$ examples.

Furthermore, we show that for every $(x, y) \in S$ the list $\mu(x)$ is indeed of size at most $k - 1$. This is true since for every $x \in \mathcal{X}$ by the definition of $H$ it holds that $\sum_{\ell \in \mathcal{Y}} H(x, \ell) \leq T$, and so we have that for all $(x, y) \in S$ it holds that at most $k - 1$ labels $\ell \in \mathcal{Y}$ can satisfy $H(x, \ell) > T/k$, and thus $|\mu(x)| \leq k - 1$.

Lastly, we observe that $\mu$ is consistent with $S$. That is, by Equation (20) and the definition of $\mu$ we know that for every $(x, y) \in S$, we have $y \in \mu(x)$.

Overall, we get that with probability at least $1 - T \cdot \delta/(2T) = 1 - \delta/2$ over the random choice of $Tm_0$ examples, it holds that the list function $\mu$ is consistent with the sample $S$. Moreover, it can be represented using only $r$ examples, where $r = Tm_0$.

We can now apply a sample compression scheme bound to obtain the final result. Specifically, we consider the bound given in Theorem 7 for a $m \to r$ sample compression scheme for $k$-lists. Then, if $\mu$ is consistent with $S$ then with probability of at least $1 - \delta/2$ over the choice of $S$ it also holds that,

$$\Pr_{(x,y)\sim\mathcal{D}} \left[ y \notin \mu(x) \right] \leq \frac{r \ln(m) + \ln(2/\delta)}{m - r}.$$

Plugging in $r = Tm_0$ from above, and setting

$$m = \frac{8 \, m_0 \ln^2(m)}{\sigma^2 \epsilon} + \frac{\ln(2/\delta)}{\epsilon} + \frac{8 \ln(m) m_0}{\sigma^2},$$

yields the desired bound. That is, with probability at least $1 - \delta$, we have $\Pr_{\mathcal{D}}[y \notin \mu(x)] \leq \epsilon$. $\qquad\square$

## B.1 Proof of Lemma 3

*Proof of Lemma 3.* First, recall that the given oracle is such that for every $\mathcal{H}$-realizable distribution $\mathcal{D}$, when given $S \sim \mathcal{D}^{m_\ell}$ for $m_\ell \geq m_\ell(\epsilon, \delta)$, it returns $\mu_S : \mathcal{X} \to \mathcal{Y}^k$ such that with probability $1 - \delta$,

$$\Pr_{(x,y)\sim\mathcal{D}} \left[ y \in \mu_S(x) \right] \geq 1 - \epsilon. \tag{21}$$

Then, we show that there is a $\gamma$-weak learning algorithm $\mathcal{W}$ for $\mathcal{H}$ where $\gamma := \frac{1-2\epsilon}{k}$, for any $\epsilon > 0$. The construction of the algorithm $\mathcal{W}$ is as follows. First, let $m_\ell := m_\ell(\epsilon, \delta_0)$ for $\delta_0 = 1/2$. The $\gamma$-weak learner $\mathcal{W}$ has a sample size $m$ to be determined later such that $m \geq m_\ell$, and is given a training set $S \sim \mathcal{D}^m$. Then, the learner $\mathcal{W}$ first calls the $k$-list learner with parameters $\epsilon, \delta_0, m_\ell$ over $S$ and obtains the list function $\mu_S$ that satisfies Equation (21) with probability at least $1 - \delta_0 = 1/2$.

Henceforth, we only consider the probability-half event in which we have that Equation (21) holds. Furthermore, let $Z$ denote the set of all elements $(x, y) \sim \mathcal{D}$ for which $y \in \mu_S(x)$. Then, notice that by randomly picking $j \leq k$ and denoting the hypothesis $h_S^j : \mathcal{X} \mapsto \mathcal{Y}$ defined by the $j$-the entry of $\mu_S$, i.e., $h_S^j(x) := \mu_S(x)_j$, we get for any $(x, y) \in Z$,

$$\Pr_{j\sim\text{Unif}(k)} \left[ y = h_S^j(x) \right] \geq \frac{1}{k}.$$

Then, by von Neumann's minimax theorem [27] it holds that for any distribution $D'$ over $Z$, there exists some particular $j_0 \leq k$ for which

$$\Pr_{(x,y)\sim D'} \left[ y = h_S^{j_0}(x) \right] \geq \frac{1}{k}.$$

Since this is true for the distribution $\mathcal{D}$ conditioned on $Z$ as well, the algorithm can pick $j \leq k$ uniformly at random and so it will pick $j_0$ with probability $\frac{1}{k}$. Overall we get that when given $S$ the

algorithm can call the list learner to obtain $\mu_S$ and sample $j$ uniformly to obtain $h_S^j : \mathcal{X} \mapsto \mathcal{Y}$ such that,

$$\Pr_{S \sim \mathcal{D}^{m_\ell}}\left[ \Pr_{j \sim \text{Unif}(k)}\left[ \Pr_{(x,y) \sim \mathcal{D}}\left[y = h_S^j(x)\right] \geq \frac{1 - \epsilon}{k} \right] \geq \frac{1}{k} \right] \geq 1 - \delta_0.$$

Since we set $\delta_0 = 1/2$, we get that,

$$\Pr_{S \sim \mathcal{D}^{m_\ell}, j \sim \text{Unif}(k)}\left[ \Pr_{(x,y) \sim \mathcal{D}}\left[y = h_S^j(x)\right] \geq \frac{1 - \epsilon}{k} \right] \geq \frac{1}{2k}. \tag{22}$$

Lastly, note that the confidence parameter of $\frac{1}{2k}$ can easily be made arbitrarily large using standard confidence-boosting techniques. That is, for any $\delta > 0$, we can set $q = 2k \log(2/\delta)$ and draw $q$ sample sets $S_1, ..., S_q$ from $\mathcal{D}^{m_\ell}$ and call the algorithm we described so far to generate hypotheses $h_1, ..., h_q$ which satisfy Equation (22) with respect to the different samples $S$, and different random draws of $j \sim \text{Unif}(k)$. By the choice of $q$, we have that the probability that no $h$ has accuracy at least $\frac{1-\epsilon}{k}$, is at most $(1 - \frac{1}{2k})^q \leq e^{-\frac{q}{2k}} = \delta/2$. Thus, we get that with probability at least $1 - \delta/2$ at least one of these hypothesis is good, in that it has accuracy of at least $\frac{1-\epsilon}{k}$. We then pick the best hypothesis as tested over another independent set $S_{q+1}$, which we sample i.i.d. from $\mathcal{D}^r$ where $r = \frac{10 \log(2q/\delta)}{\epsilon'^2}$ and $\epsilon' = \epsilon/k$. Then, by Chernoff's inequality we have for each of the $h_1, ..., h_q$ that with probability $1 - \delta/(2q)$ that the gap between its accuracy over $S_{q+1}$ and over $\mathcal{D}$ is at most $\epsilon'/2$. By taking the union bound, we get that this holds with probability $1 - \delta/2$ for all $h_1, ..., h_q$ simultaneously.

Thus when we choose the $h$ with the most empirical accuracy over $S_{q+1}$, then with probability $1 - \delta$ it will have an empirical accuracy of at least $\frac{1-\epsilon}{k} - \epsilon'/2$, and true accuracy of at least $\frac{1-\epsilon}{k} - \epsilon'$. Therefore, with probability at least $1 - \delta$ we get that the hypothesis chosen by our final algorithm has accuracy of at least $\gamma = \frac{1-2\epsilon}{k}$ over $\mathcal{D}$.

Overall, we get that $\mathcal{W}$ is a $\gamma$-weak PAC learner for $\mathcal{H}$, where $\gamma = \frac{1-2\epsilon}{k}$, and with sample complexity $m = O\left( k \cdot \log(1/\delta) \cdot m_\ell(\epsilon, 1/2) + k^2 \cdot \frac{\log(k/\delta)}{\epsilon^2} \right)$. $\qquad \square$

## B.2 Proof of Theorem 3

*Proof of Theorem 3.* We start with showing that $\gamma(\mathcal{H}) \geq 1/k(\mathcal{H})$. By assumption, we have that for any fixed $\epsilon_1, \delta_1 > 0$, when given $m_1 := m_1(\epsilon_1, \delta_1)$ examples $S$ drawn i.i.d. from a $\mathcal{H}$-realizable distribution $\mathcal{D}$, the list learner returns a function $\mu_S : \mathcal{X} \to \mathcal{Y}^{k(\mathcal{H})}$ such that,

$$\Pr_{S \sim \mathcal{D}^{m_1}}\left[ \Pr_{(x,y) \sim \mathcal{D}}\left[y \in \mu_S(x)\right] \geq 1 - \epsilon_1 \right] \geq 1 - \delta_1. \tag{23}$$

By Lemma 3 we get that for any $\epsilon > 0$, there is a $\gamma$-weak learner for $\mathcal{H}$, where $\gamma := \frac{1-2\epsilon}{k(\mathcal{H})}$.

Moreover, by definition of $\gamma(\mathcal{H})$ as the supremum over all such $\gamma$ values, we have $\gamma(\mathcal{H}) \geq \frac{(1-\epsilon)}{k(\mathcal{H})}$. Lastly, since this holds for any choice of $\epsilon$, we can take $\epsilon \to 0$ and get $\gamma(\mathcal{H}) \geq \frac{1}{k(\mathcal{H})}$.

We now show that $\gamma(\mathcal{H}) \leq 1/k(\mathcal{H})$. Let $k$ denote the smallest integer for which $\frac{1}{k} < \gamma(\mathcal{H})$, and let $\sigma_0 := \gamma(\mathcal{H}) - 1/k > 0$. We will now show how to construct a list learner via the given weak learner, for a list size of $k - 1$. It suffices to show that, since by the choice of $k$ we have $\gamma(\mathcal{H}) \leq 1/(k-1)$, and so it holds that $k(\mathcal{H}) \leq k - 1 \leq 1/\gamma(\mathcal{H})$ and we get the desired bound of $\gamma(\mathcal{H}) \leq 1/k(\mathcal{H})$. Thus, all that is left is to show the existence of the $(k-1)$-list learner.

Let $\epsilon_1, \delta_1 > 0$, and let $m(\epsilon_1, \delta_1)$ be determined as $m$ in lemma 5 for $\epsilon_1, \delta_1$. Set $T = \frac{8 \log(m)}{\sigma_0^2}$. Let $m_0 := m_0(\delta_0 := \delta_1/(2T))$ denote the sample size of a $\gamma$-weak PAC learner with $\gamma := \frac{\sigma_0}{2} + \frac{1}{k}$. Denote this weak learner by $\mathcal{W}$. Then, by applying Lemma 5 with the $\gamma$-weak learner $\mathcal{W}$ and with $\sigma := \sigma_0$, we obtain the $(k-1)$-list learner, which concludes the proof.

$\square$

### B.3 Proof of Theorem 4

*Proof of Theorem 4.* The proof follows by constructing a weak learner from the given oracle, and then applying a weak-to-list boosting procedure.

First, by Lemma 3 we get that there is a $\gamma$-weak learning algorithm $\mathcal{W}$ for $\mathcal{H}$ where $\gamma := \frac{1-2\epsilon_0}{k_0}$.[5]

Then, by applying lemma 5 with this $\gamma$-weak learner $\mathcal{W}$, we obtain a $(t-1)$-List PAC learner, where $\frac{1}{t} < \gamma \leq \frac{1}{t-1}$. Then, we also get that $\frac{1-2\epsilon_0}{k_0} \leq \frac{1}{t-1}$ and so $(t-1) \leq \left\lfloor \frac{k_0}{1-2\epsilon_0} \right\rfloor$.

Therefore, there is a $k$-List PAC learning algorithm for $\mathcal{H}$ with a *fixed* list size $k = \left\lfloor \frac{k_0}{1-2\epsilon_0} \right\rfloor$. ☐

## C Proof of Theorem 5

At a high level, the proof describes a construction of a simple weak learner which is shown to satisfy our BRG condition (Definition 4). This implies it is also amenable to our boosting method, which yields the final result given in Theorem 5.

The construction of the weak learner is based on an object called the *One-inclusion Graph* (OIG) [13, 20] of a hypothesis class, which is often useful in devising learning algorithms. Typically, one is interested in *orienting the edges* of this graph in a way that results in accurate learning algorithms. As in the analysis used to characterize PAC, and List PAC learning [5, 8], as well as in most applications of the OIG [9, 13, 20], a good learner corresponds to an orientation with a small maximal out-degree. However, here we show that a simpler task of minimizing the in-degree is sufficient to obtain a reasonable weak learner, and therefore a strong learner as well.

We start with introducing relevant definitions.

The DS dimension was originally introduced by [9]. Here we follow the formulation given in [5], and so we first introduce the notion of *pseudo-cubes*.

**Definition 9** (Pseudo-cube). *A class $\mathcal{H} \subseteq \mathcal{Y}^d$ is called a* pseudo-cube *of dimension $d$ if it is non-empty, finite and for every $h \in \mathcal{H}$ and $i \in [d]$, there is an $i$-neighbor $g \in \mathcal{H}$ of $h$ (i.e., $g(i) \neq h(i)$ and $g(j) = h(j)$ for all $j \neq i$).*

When $\mathcal{Y} = \{0, 1\}$, the two notions "Boolean cube" and "pseudo-cube" coincide: The Boolean cube $\{0, 1\}^d$ is of course a pseudo-cube. Conversely, every pseudo-cube $\mathcal{H} \subseteq \{0, 1\}^d$ is the entire Boolean cube $\mathcal{H} = \{0, 1\}^d$. When $|\mathcal{Y}| > 2$, the two notions do not longer coincide. Every copy of the Boolean cube is a pseudo-cube, but there are pseudo-cubes that are not Boolean cubes (see [5] for further details). We are now ready to define the DS dimension.

**Definition 10** (DS dimension). *A set $S \in \mathcal{X}^n$ is DS-shattered by $\mathcal{H} \subseteq \mathcal{Y}^\mathcal{X}$ if $\mathcal{H}|_S$ contains an $n$-dimensional pseudo-cube. The DS dimension $d_{DS}(\mathcal{H})$ is the maximum size of a DS-shattered sequence.*

A natural analogue of this dimension in the context of predicting lists of size $k$ is as follows:

**Definition 11** ($k$-DS dimension [8]). *Let $\mathcal{H} \subseteq \mathcal{Y}^\mathcal{X}$ be a hypothesis class and let $S \in \mathcal{X}^d$ be a sequence. We say that $\mathcal{H}$ $k$-DS shatters $S$ if there exists $\mathcal{F} \subseteq \mathcal{H}, |\mathcal{F}| < \infty$ such that $\forall f \in \mathcal{F}|_S, \forall i \in [d]$, $f$ has at least $k$ $i$-neighbors. The $k$-DS dimension of $\mathcal{H}$, denoted as $d_{DS}^k = d_{DS}^k(\mathcal{H})$, is the largest integer $d$ such that $\mathcal{H}$ $k$-DS shatters some sequence $S \in \mathcal{X}^d$.*

Observe that the two definitions above differ only in the number of $i$-neighbors they require each hypothesis to have, and that $d_{DS} = d_{DS}^k$ for $k = 1$. Moreover, there are simple hypothesis classes with infinite DS dimension, yet finite $k$-DS dimension for $k \geq 2$, see Example 3 in [8].

Next, we define the object called the *One-inclusion Graph* (OIG) [13, 20] of a hypothesis class, and related definitions which will be used for the construction of the weak learner.

**Definition 12** (One-inclusion Graph [13, 20]). *The one-inclusion graph of $\mathcal{H} \subseteq \mathcal{Y}^n$ is a hypergraph $\mathcal{G}(\mathcal{H}) = (V, E)$ that is defined as follows.[6] The vertex-set is $V = \mathcal{H}$. For each $i \in [n]$ and*

---

[5]We note that although Lemma 3 assumes access to a $k$-List PAC learner, the same argument holds for the weaker version of a list learner with list size $k_0$ which depends on $\epsilon_0$, and the proofs of both cases is identical.

[6]We use the term "one-inclusion graph" although it is actually a hypergraph.

$f : [n] \setminus \{i\} \to \mathcal{Y}$, let $e_{i,f}$ be the set of all $h \in \mathcal{H}$ that agree with $f$ on $[n] \setminus \{i\}$. The edge-set is

$$E = \{e_{i,f} : i \in [n], f : [n] \setminus \{i\} \to \mathcal{Y}, e_{i,f} \neq \emptyset\}. \tag{24}$$

*We say that the edge $e_{i,f} \in E$ is in the direction $i$, and is adjacent to/contains the vertex $v$ if $v \in e_{i,f}$. Every vertex $h \in V$ is adjacent to exactly $m$ edges. The size of the edge $e_{i,f}$ is the size of the set $|e_{i,f}|$.*

We remark that similarly to [5, 8], edges could be of size one, and each vertex $v$ is contained in exactly $n$ edges. This is not the standard structure of edges in hypergraphs, but we use this notation because it provides a better model for learning problems.

With respect to the one-inclusion graph, one can think about the *degrees* of its vertices, all of which originally introduced in [8], and are natural generalizations of the degrees defined in [9, 5]

**Definition 13** ($k$-degree). *Let $\mathcal{G}(\mathcal{H}) = (V, E)$ be the one-inclusion graph of $\mathcal{H} \subseteq \mathcal{Y}^m$. The $k$-degree of a vertex $v \in V$ is*

$$\deg^k(v) = |\{e \in E : v \in e, |e| > k\}|.$$

We now define the notion of *orienting* edges of a one-inclusion graph to lists of vertices they are adjacent to. As alluded to earlier, an orientation corresponds to the behavior of a (deterministic) learning algorithm while making predictions on an unlabeled test point, given a set of labeled points as input.

**Definition 14** (List orientation). *A list orientation $\sigma^k$ of the one-inclusion graph $\mathcal{G}(\mathcal{H}) = (V, E)$ having list size $k$ is a mapping $\sigma^k : E \to \{V' \subseteq V : |V'| \leq k\}$ such that for each edge $e \in E$, $\sigma^k(e) \subseteq e$.*

The $k$-out-degree of a list orientation $\sigma^k$ is defined as:

**Definition 15** ($k$-out-degree of a list orientation). *Let $\mathcal{G}(\mathcal{H}) = (V, E)$ be the one-inclusion graph of a hypothesis class $\mathcal{H}$, and let $\sigma^k$ be a $k$-list orientation of it. The $k$-out-degree of $v \in V$ in $\sigma^k$ is*

$$\mathsf{outdeg}^k(v; \sigma^k) = |\{e : v \in e, v \notin \sigma^k(e)\}|. \tag{25}$$

*The maximum $k$-out-degree of $\sigma^k$ is*

$$\mathsf{outdeg}^k(\sigma^k) = \sup_{v \in V} \mathsf{outdeg}^k(v; \sigma^k). \tag{26}$$

The following lemmas demonstrates how a bound on the dimensions helps one greedily construct a list orientation of small maximum $k$-out-degree.

**Lemma 6** (Lemma 3.1, [8]). *If $\mathcal{H} \subseteq \mathcal{Y}^{d+1}$ has $k$-DS dimension at most $d$, then there exists a $k$-list orientation $\sigma^k$ of $\mathcal{G}(\mathcal{H})$ with $\mathsf{outdeg}^k(\sigma^k) \leq d$.*

We now describe a list version of the one-inclusion algorithm below, originally introduced by [8].

---

**Algorithm 5** The one-inclusion list algorithm for $\mathcal{H} \subseteq \mathcal{Y}^{\mathcal{X}}$

---

**Input:** An $\mathcal{H}$-realizable sample $U = ((x_1, y_1), \ldots, (x_n, y_n))$.
**Output:** A $k$-list hypothesis $\mu_U^k : \mathcal{X} \to \{Y \subseteq \mathcal{Y} : |Y| \leq k\}$.

For each $x \in \mathcal{X}$, the $k$-list $\mu_S^k(x)$ is computed as follows:
1: Consider the class of all patterns over the *unlabeled data* $\mathcal{H}|_{(x_1,\ldots,x_n,x)} \subseteq \mathcal{Y}^{n+1}$.
2: Find a $k$-list orientation $\sigma^k$ of $\mathcal{G}(\mathcal{H}|_{(x_1,\ldots,x_n,x)})$ that *minimizes* the *maximum* $k$-out-degree.
3: Consider the following edge defined by revealing all labels in $U$:

$$e = \{h \in \mathcal{H}|_{(x_1,\ldots,x_n,x)} : \forall i \in [n] \ h(i) = y_i\}.$$

4: Set $\mu_U^k(x) = \{h(x) : h \in \sigma^k(e)\}$.

---

**Lemma 7.** *Let $\mathcal{H} \subseteq \mathcal{Y}^{\mathcal{X}}$ be a hypothesis class, let $\mathcal{D}$ be an $\mathcal{H}$-realizable distribution over $\mathcal{X} \times \mathcal{Y}$, and let $k, n > 0$ be integers. Let $M$ be an upper bound on the maximum $k$-out-degree of all orientations $\sigma^k$ chosen by Algorithm 5, and let $\mu_U^k$ denote its output for an input sample $U \sim \mathcal{D}^n$. Then,*

$$\Pr_{(U,(x,y)) \sim \mathcal{D}^{n+1}} \left[ \mu_U^k(x) \not\ni y \right] \leq \frac{M}{n+1}.$$

*Proof.* By the leave-one-out symmetrization argument (Fact 14, [7]) we have,

$$\Pr_{(U,(x,y)) \sim \mathcal{D}^{n+1}} \left[ \mu_U^k(x) \not\ni y \right] = \Pr_{(U',i) \sim \mathcal{D}^{n+1} \times \mathrm{Unif}(n+1)} \left[ \mu_{U'_{-i}}^k(x_i') \not\ni y_i' \right].$$

It therefore suffices to show that for every sample $U'$ that is realizable by $\mathcal{H}$,

$$\Pr_{i \sim \mathrm{Unif}(n+1)} \left[ \mu_{U'_{-i}}^k(x_i') \not\ni y_i' \right] \leq \frac{M}{n+1}. \tag{27}$$

Fix $U'$ that is realizable by $\mathcal{H}$ for the rest of the proof. Denote by $\sigma^k$ the orientation of $\mathcal{G}(\mathcal{H}')$ that the algorithm chooses.

Let $y'$ denote the vertex in $\mathcal{G}(\mathcal{H}')$ defined by $y' = (y_1', \ldots, y_{n+1}')$, and let $e_i$ denote the edge in the $i^{\text{th}}$ direction adjacent to $y'$. Then, we have that

$$\begin{aligned}
\Pr_{i \sim \mathrm{Unif}(n+1)} \left[ \mu_{U'_{-i}}^k(x_i') \not\ni y_i' \right] &= \frac{1}{n+1} \sum_{i=1}^{n+1} \mathbb{1} \left[ \mu_{U'_{-i}}^k(x_i') \not\ni y_i' \right] \\
&= \frac{1}{n+1} \sum_{i=1}^{n+1} \mathbb{1} \left[ \sigma^k(e_i) \not\ni y' \right] \\
&= \frac{\mathrm{outdeg}^k(y'; \sigma^k)}{n+1} \\
&\leq \frac{M}{n+1}.
\end{aligned}$$

$\square$

## C.1 Initial phase: pre-processing

**Lemma 8.** *Let $\mathcal{H} \subseteq \mathcal{Y}^{\mathcal{X}}$ be a hypothesis class with $k$-DS dimension $d < \infty$. Then, for every $\mathcal{H}$-realizable set $S \in (\mathcal{X} \times \mathcal{Y})^m$, any $\mathcal{D}$ distribution over $S$, for a sample $U \sim \mathcal{D}^n$ for $n := d$, when given $U \in (\mathcal{X} \times \mathcal{Y})^n$, Algorithm 5 returns $h_U : \mathcal{X} \mapsto \mathcal{Y}$ such that,*

$$\mathbb{E}_{U \sim \mathcal{D}^d} \left[ \Pr_{(x,y) \sim \mathcal{D}} [y \in \mu_U^k(x)] \right] \geq \frac{1}{d+1}. \tag{28}$$

*Proof.* This follows directly by combining Lemma 6 and Lemma 7 $\square$

**Lemma 9.** *Let $\mathcal{H} \subseteq \mathcal{Y}^{\mathcal{X}}$ with $k$-DS dimension $d < \infty$, and let $S \in (\mathcal{X} \times \mathcal{Y})^m$ be an $\mathcal{H}$-realizable set. Denote by $\mathcal{L}$ a $k$-list learning algorithm as given in Algorithm 5. Denote $p = k \cdot q$ and $q = (d+1) \ln(2m)$, and let $r = qd = O(d^2 \ln(m))$. Then, there exists an algorithm $\mathcal{A}$ based on an $m \to r$ compression scheme for $p$-lists as described next. When provided with $S$, and given oracle access to $\mathcal{L}$, the algorithm yields a list function $\mu : \mathcal{X} \to \mathcal{Y}^p$ such that $\mu$ is consistent with $S$.*

*Proof.* We describe an algorithm that sequentially finds $q$ subsets of examples $S_1', \ldots, S_q' \subseteq S$ each of size $d$, and returns a list $\mu : \mathcal{X} \mapsto \mathcal{Y}^k$, using $r$ examples. First, let $\mathcal{U}$ denote the uniform distribution over the $m$ examples in $S$ and let $\alpha = \frac{1}{d+1}$. By Lemma 8 applied to the distribution $\mathcal{U}$, for a random sample $S_1' \sim \mathcal{U}^d$, in expectation at least $\alpha$ of the examples $(x,y) \in S$ satisfy that $y \in \mu_1(x)$, where $\mu_1$ is the output of $\mathcal{L}$ over $S_1'$. In particular, there exists $S_1'$ for which the corresponding $\mu_1$ covers at least $\alpha m$ examples. Algorithmically, it can be found via a brute-force search over all possible $m^d$ subsets $S_1' \subseteq S$. Next, remove from $S$ all examples $(x,y)$ for which $y \in \mu_1(x)$ and repeat the same reasoning on the remaining sample. This way at each step $j$ we find a sample $S_j'$ and a list

$\mu_j = \mathcal{L}(S_j')$ that covers at least an $\alpha$-fraction of the remaining examples. After $q$ steps, all examples in S are covered since $|S_q| \leq (1 - \frac{1}{d+1})^q m < e^{-\frac{q}{d+1}} m < 1$.

The final list returned by our method is $\mu$ defined by the concatenation:

$$\mu(x) = \bigcup_{j=1}^{q} \mu_j(x),$$

and is based only on the examples in $S_q', ..., S_q'$ which is a total of $r$ examples as claimed.

$\square$

## C.2   Constructing a weak learner for *wrong* labels

The following lemmas demonstrates how a bound on the dimensions helps one greedily construct a list orientation of small maximum $(p-1)$-out-degree, for a class with $p$ labels.

**Lemma 10.** *Let $p, n \in \mathbb{N}$ and assume $\mathcal{H} \subseteq [p]^n$ is a hypothesis class with a $(p-1)$-DS dimension at most $d$, where $d < n$. Then, there exists an orientation $\sigma^{p-1}$ of $\mathcal{G}(\mathcal{H})$ with $\mathsf{outdeg}^{p-1}(\sigma^{p-1}) \leq 4(p-1)^2 d$.*

*Proof.* Notice that $\mathcal{H}$ is finite, and so the number of vertices in $\mathcal{G}(\mathcal{H})$ is finite. We consider another notion of dimension called the *k-Exponential dimension*. We say that $S \subseteq \mathcal{X}$ is $k$-exponential shattered by $\mathcal{H} \subseteq \mathcal{Y}^{\mathcal{X}}$ if $|\mathcal{H}|_S| \geq (k+1)^{|S|}$. The $k$-exponential dimension $d_E^k(\mathcal{H})$ is the maximum size of an $k$-exponential shattered sequence.

Notice that for the case that $\mathcal{Y} = [p]$ and $k = p - 1$, we have that a set $S$ is $(p-1)$-exponential shattered if $|\mathcal{H}|_S| \geq p^{|S|}$. Observe that this definition coincides with that of the $(p-1)$-DS dimension shattering. That is, a set $S$ is $(p-1)$-DS shattered if for every $h \in \mathcal{H}|_S$ and every $i \in S$ it holds that $h$ has exactly $p-1$ neighbors that agree with it on $[n] \setminus \{i\}$. This definition implies that $|\mathcal{H}|_S| \geq p^{|S|}$.

Therefore, for $\mathcal{H} \subseteq [p]^n$ we have that

$$d_{DS}^{p-1}(\mathcal{H}) = d_E^{p-1}(\mathcal{H}).$$

Then, by Corollary 6.5 of [8] we get that there is an there is a $(p-1)$-list orientation of $\mathcal{G}(\mathcal{H})$ with maximum $(p-1)$-out-degree at most $4(p-1)^2 d_E^{p-1}(\mathcal{H})$. Replacing $d_E^{p-1}(\mathcal{H})$ by $d_{DS}^{p-1}(\mathcal{H})$ yields the desired bound. $\square$

The following lemma is analogous to Lemma 8.

**Lemma 11.** *Let $p \in \mathbb{N}$ and assume $\mathcal{H} \subseteq [p]^{\mathcal{X}}$ is a hypothesis class with a $(p-1)$-DS dimension at most $d$. For every $\mathcal{H}$-realizable set $S \in (\mathcal{X} \times [p])^m$, any $\mathcal{D}$ distribution over $S$, denote by $\mu_U^{p-1} : \mathcal{X} \mapsto [p]^{p-1}$ the $p-1$-list that is the output of Algorithm 5 when given a sample $U \sim \mathcal{D}^{4pd}$. Then,*

$$\mathbb{E}_{U \sim \mathcal{D}^{4pd}} \left[ \Pr_{(x,y) \sim \mathcal{D}} [y \in \mu_U^{p-1}(x)] \right] \geq 1 - \frac{1}{4p}. \tag{29}$$

*Proof.* For any $U = \{(x_1, y_1), ..., (x_n, y_n)\}$ where $n = 4pd$, and any text point $(x, y) \sim \mathcal{D}$, let $H \subseteq [p]^{n+1}$ be the class of all patterns over the unlabeled data $\mathcal{H}|_{(x_1, ..., x_n, x)}$, as in Algorithm 5. Notice that by Lemma 10, we have that for any such class $H$, there is an orientation with a maximal $(p-1)$-out-degree of at most $d$. Therefore, by Lemma 7 we get that,

$$\mathbb{E}_{U \sim \mathcal{D}^{4pd}} \left[ \Pr_{(x,y) \sim \mathcal{D}} [y \in \mu_U^{p-1}(x)] \right] = \Pr_{(U, (x,y)) \sim \mathcal{D}^{4pd+1}} \left[ y \in \mu_U^{p-1}(x) \right]$$

$$\geq 1 - \frac{d}{4pd + 1} = \frac{d(4p - 1) + 1}{4pd + 1} > (4p - 1)/4p.$$

$\square$

**Theorem 8.** *Let $p \in \mathbb{N}$ and assume $\mathcal{H} \subseteq [p]^{\mathcal{X}}$ is a hypothesis class with a $(p-1)$-DS dimension at most $d$, and let $S \in (\mathcal{X} \times [p])^m$ be an $\mathcal{H}$-realizable set of examples. Denote by $\mathcal{L}$ a $(p-1)$-list learning algorithm as given in Algorithm 5, and let $r = 4pd \cdot \lceil 8p^2 \ln(2m) \rceil$.*

*Then, there exists an algorithm $\mathcal{A}$ based on an $m \rightarrow r$ compression scheme for $(p-1)$-lists such that, when provided with $S$ and given oracle access to $\mathcal{L}$, the algorithm $\mathcal{A}$ yields a list function $\mu : \mathcal{X} \rightarrow [p]^{p-1}$ that is consistent with $S$.*

*Proof.* Let $D$ denote any distribution over the $m$ examples in $S$. For a random sample $U \sim \mathcal{U}^{4pd}$ denote $\mu_U = \mathcal{L}(U)$. By Lemma 11 we have that in expectation over the random choice of $U$, it holds that $\Pr_{(x,y)\sim D}[y \in \mu_U(x)] \geq 1 - \frac{1}{4p}$. Therefore, there exists a *particular* set $U$ of size $4pd$ for which this holds as well. Algorithmically, this particular set can be found via a brute-force search over all possible $m^{4pd}$ subsets $U \subseteq S$ of size $4pd$. We now have a $(p-1)$-list function $\mu_U = \mathcal{L}(U)$ such that with probability 1 satisfies $\Pr_{(x,y)\sim D}[y \in \mu_U(x)] \geq 1 - \frac{1}{4p}$.

Notice that while $\mu_U$ covers a large fraction of the examples, it is not entirely consistent with $S$. The remainder of the proof will be concerned with boosting the list learner we have described above, into a new list $\mu : \mathcal{X} \rightarrow \mathcal{Y}^{p-1}$ that is indeed consistent with the entire sample.

Towards that end, we first use $\mu_U$ to define a classifier $f_U : \mathcal{X} \mapsto [p]$ that predicts the *incorrect* labels. Specifically, we set $f_U(x) = \hat{y}$ for $\hat{y} \in [p] \setminus \mu_U(x)$. Thus, we have that with probability 1,

$$\Pr_{(x,y)\sim D}[f_U(x) = y] < \frac{1}{4p}. \tag{30}$$

So far we have shown that for *any* distribution $D$ over $S$ there is a set $U \in S^{4pd}$ and a corresponding function $f_U : \mathcal{X} \mapsto [p]$ (constructed from $\mu_U = \mathcal{L}(U)$ as above), that satisfies Equation (30).

Next, by von Neumann's minimax theorem [27] there exists a distribution $Q$ over all possible subsets $U \subseteq S$ of size at most $4pd$ such that for all examples $(x, y) \in S$ it holds that:

$$\Pr_{U \sim Q}[f_U(x) = y] < \frac{1}{4p}.$$

Let $U_1, ..., U_\ell$ be random samples from $Q$ where $\ell = \lceil 8p^2 \ln(2m) \rceil$. Denote $\bar{U}_\ell = (U_1, ..., U_\ell)$, and define $F_{\bar{U}_\ell}$ to be a corresponding averaged-vote, such that for any $x \in \mathcal{X}$ and $y \in [p]$:

$$F_{\bar{U}_\ell}(x, y) = \frac{1}{\ell} \sum_{j=1}^{\ell} \mathbb{1}\Big[f_{U_j}(x) = y\Big].$$

Observe that for a fixed example pair $(x, y) \in S$ then by a Chernoff bound we have,

$$\Pr_{\bar{U}_\ell \sim Q^\ell}\left[F_{\bar{U}_\ell}(x, y) \geq \frac{1}{2p}\right] \leq \Pr_{\bar{U}_\ell \sim Q^\ell}\left[F_{\bar{U}_\ell}(x, y) \geq \mathbb{E}[F_{\bar{U}_\ell}(x, y)] + \frac{1}{4p}\right] \tag{31}$$

$$\leq e^{-2(\ell/4p)^2/\ell} = e^{-\ell/(8p^2)} \leq \frac{1}{2m}. \tag{32}$$

Then, by a union bound over all $m$ examples in $S$ we have that with positive probability over the random choice of $\bar{U}_\ell$ it holds that $F_{\bar{U}_\ell}(x, y) < \frac{1}{2p}$ over all $(x, y) \in S$ simultaneously. This implies that there exist a particular choice of $\bar{U}_\ell$ for which we have with probability 1 for all $(x, y) \in S$ that:

$$y \notin \arg\max_{\hat{y} \in [p]} F_{\bar{U}_\ell}(x, \hat{y}).$$

In words, this means that taking the plurality-vote as induced by $\bar{U}_\ell$, we will yield an *incorrect* label with probability 1 over all examples in $S$ simultaneously.

Lastly, we are ready to define the final list function $\mu : \mathcal{X} \rightarrow [p]^{p-1}$ by:

$$\mu(x) = [p] \setminus \arg\max_{\hat{y} \in [p]} F_{\bar{U}_\ell}(x, \hat{y}).$$

To conclude, we have constructed an algorithm based on $m \rightarrow 4pd \cdot \ell$ compression scheme for $(p-1)$-lists, that is consistent with the training sample $S$, as claimed. $\qquad\square$

**Algorithm 6** $k$-List PAC Learning for a class $\mathcal{H} \subseteq \mathcal{Y}^{\mathcal{X}}$ with $d_{DS}^k = d < \infty$

---

**Given:** Training data $S \in (\mathcal{X} \times \mathcal{Y})^m$.
**Output:** A list $\mu : \mathcal{X} \mapsto \mathcal{Y}^k$.

1: Set $p = k \cdot (d+1) \cdot \ln(2m)$ (as chosen in Lemma 9).
2: Initialize: applying pre-processing as given in Lemma 9 to get $\mu_1 : \mathcal{X} \mapsto \mathcal{Y}^p$.
3: **for** $j = 1, ..., p - k$ **do**
4:     Let $S^j$ and $\mathcal{H}^j$ denote $S$ and $\mathcal{H}$ with all labels converted from $\mathcal{Y}$ to $[p - j + 1]$ via $\mu_j$.
5:     Call the list learner given in Theorem 8 over $S^j$ and $\mathcal{H}^j$, to obtain $\tilde{\mu}_j : \mathcal{X} \to [p - j + 1]^{p-j}$.
6:     Define $\mu_{j+1} : \mathcal{X} \to \mathcal{Y}^{p-j}$ by :

$$\mu_{j+1}(x) = \Big\{ y : \exists \ell, \ \mu_j(x)_\ell = y \ \wedge \ \ell \in \tilde{\mu}_j(x) \Big\}.$$

7: **end for**
8: Output the final list $\mu := \mu_{p-k+1}$.

---

*Proof of Theorem 5.* The proof is given via a sample compression scheme argument, demonstrating that for a class with a finite $k$-DS dimension $d$, Algorithm 6 is a $k$-list PAC learner. That is, we prove that it returns a list $\mu : \mathcal{X} \mapsto \mathcal{Y}^k$ which can be represented using a small number of training examples, and that it is consistent with the entire training set.

We will then show for each $j = 1...p - k$ that $\mu_{j+1}$ satisfies the following 2 properties: (a) the list $\mu_{j+1}$ is consistent with the entire sample $S$, and (b) it can be represented using only a small number $r_j$ of training examples.

First, notice that by Lemma 9 it is guaranteed $\mu_1$ is consistent with $S$, and that it can be represented by $r_1 = d(d+1)\ln(2m)$ examples, as it is constructed by an $m \to r_1$ compression scheme.

Next, we will show that the 2 properties (a) and (b) above hold for all $j \geq 2$. Notice that in each round $j$ of Algorithm 6, the class $\mathcal{H}^j$ has only $p - j + 1$ labels, and its $k$-DS dimension remains $d$ as in $\mathcal{H}$. Furthermore, by the properties of the $k$-DS dimension it holds that for any $k' > k$ we have $d_{DS}^{k'} \leq d_{DS}^k$. Thus, for all $j \leq p - k$,

$$d_{DS}^{p-j}(\mathcal{H}^j) \leq d_{DS}^k(\mathcal{H}^j) = d.$$

Therefore, we can apply Theorem 8 and get that both properties (a) and (b) hold with $r_j = 4(p - j + 1)d \cdot \lceil 8(p - j + 1)^2 \ln(2m) \rceil$, for all $j = 2, ..., p - k$.

Overall, we have shown that the final list $\mu := \mu_{p-k+1}$ is both consistent with the sample $S$, and can be represented using only $r$ examples, where,

$$r = r_1 + \sum_{j=2}^{p-k+1} r_j \tag{33}$$

$$= d(d+1)\ln(2m) + \sum_{j=2}^{p-k+1} \Big( 4(p - j + 1)d \cdot \lceil 8(p - j + 1)^2 \ln(2m) \rceil \Big) \tag{34}$$

$$\leq d(d+1)\ln(2m) + 32d\ln(2m)p^4 + p \tag{35}$$

$$= O\Big( d^5 \cdot k^4 \cdot \ln^5(m) \Big), \tag{36}$$

where the bound follows by plugging in the value of $p$ from Algorithm 6 (and Lemma 9).

We can now apply a sample compression scheme bound to obtain the final result. Specifically, we apply Theorem 7, for a $m \to r$ sample compression scheme algorithm $\mathcal{A}$ equipped with a reconstruction function $\rho$ (see Definition 8). We denote $\mathrm{err}_{\mathcal{D}}(\mu) = \Pr_{(x,y)\sim\mathcal{D}}[\mu(x) \not\ni y]$. Then, by Theorem 7 we get that for any $\delta > 0$,

$$\Pr_{S\sim\mathcal{D}^m} \left[ \mathrm{err}_{\mathcal{D}}(\mu) > \frac{r\ln(m) + \ln(1/\delta)}{m - r} \right] \leq \delta,$$

Plugging in $r$ from Equation (36) yields the desired bound. $\qquad\square$

