# OpenReview forum: "Multiclass Boosting: Simple and Intuitive Weak Learning Criteria"
_NeurIPS.cc/2023/Conference — NeurIPS 2023 poster_

### Official Review · Reviewer_Ut8K · 2023-07-05

**Soundness:** 3 good
**Presentation:** 2 fair
**Contribution:** 3 good
**Rating:** 4
**Confidence:** 4

**Summary:**

This paper tackles the problem of multi-class boosting by proposing a novel definition for weak-learning based on list boosting. The authors define a weak-learning condition for multi-class learning based on the simple principle of slightly better than random guessing. This definition is then used as a stepping stone in order to filter the potential classes for a sample recursively, thus reaching a stage where either the binary weak-learning condition can be applied, or the weak-learning condition BRG leads to boostability. The authors propose several theoretical justifications and definitions for the results presented in the paper.

**Strengths:**

# Motivation
Multi-class boosting has received a lot of attention since the introduction of binary boosting, in particular since the extension of the weak-learning condition from binary to multi-class setting is tricky. As such, the novel definition of weak-learning introduced in this paper is quite interesting, especially since its simplicity mirrors the binary one.

# Theoretical results
The strongest point of this paper. The authors propose several theoretical justifications throughout the paper, and the generalization result in Theorem 2 is quite promising, same for the relation between weak PAC learning and list PAC learning in Theorem 3.

**Weaknesses:**

# Existing frameworks
There are several existing frameworks for multi-class boosting based on specific weak-learning conditions (some of which are cited in the paper). I strongly think that an actual comparison between the proposed framework and the existing ones should have been included. Particularly for methods such as Adaboost.MM and Adaboost.MR and their WL conditions, and the WL framework introduced in "Multiclass boosting and the cost of weak learning." [6].

# Experimental results
I'm not entirely sure why no experimental results are proposed in the main paper. There are several multi-class boosting that have been successfully used in practice, despite their WL conditions, as such it is important for new/novel methods to be compared to state-of-the-art approaches. In particular how the proposed method fares against state-of-the-art ones both in accuracy (or other multi-class performance measures) and in runtime. The recursive nature of Algorithm 1 and its dependance on 3 hyper-parameters might lead to prohibitive runtimes, even though several optimizations have been mentioned in the main paper.

# Conclusion and future works
I strongly think that a section on the future works and possibilities would've been more appropriate than the proof of Theorem 2. More globally, the paper could benefit from a better global organization.

**Questions:**

* How does the current framework compare to existing ones? Are the WL conditions introduced in [6] and [16] stronger than BRG?
* Any experimental results available?

**Limitations:**

The limitation of have not been discussed.

---

> ### Author Rebuttal · Authors · 2023-08-10
>
> We thank the reviewer for their feedback and suggestions.
>
> - “comparison between … existing ones” - The reviewer implies that we have not included a comparison to prior works, yet all of the works they mention are explicitly discussed in the paper. Addboost.MR is mentioned in line 130, and is a rather old extension to multiclass boosting (other works such as [16] have generalized it). Adaboost.MM is given in [16], a framework we compare with both in the introduction and in the related work section. Similarly, [6] builds off of the framework of [6] and discussed in our paper as well. To briefly explain the comparison to these results, first we note that conditions of previous works, in particular [6,16] assume the PAC setting and realizability w.r.t a hypothesis class, whereas here these restrictions are removed. Moreover, these prior works require the WL to be agnostic and learn for any re-labeling of the data. Thus, these methods operate in the rather narrow setting of assuming realizability of the data, while also having access to an agnostic learner. However, it may be interesting to examine, when restricting to the PAC setting, how these conditions are related.
> - “lack of experiments” - we emphasize that the main contribution of this work is the theoretical framework extending classical boosting theory to the multiclass setting. The algorithmic contribution is not the main focus of the paper. The question of finding the simplest and minimal assumptions on weak learnability as in the binary case has been studied in various works over the years, and in this work we provide the first intuitive and simple condition that leads to a simple boosting procedure. We leave the question of obtaining an optimal running time and performing empirical evaluations as an interesting future work. We certainly hope this will be further explored in a following line of work.

---

> > ### Comment · Reviewer_Ut8K · 2023-08-15
> >
> > Thank you for your answers. I'd like to further extend upon my previous questions, as it seems both have been misunderstood.
> >
> > Concerning the comparison to previous methods, while I do agree, that both setting have been cited in the paper, I was mostly looking for a theoretical comparison, i.e. casting the WL conditions of [6] and [16] (or other frameworks, however ancient they might be) in the present framework and study how they fare, similarly to how [16] cast Adaboost.MR in their framework and showed that the WL is too strong for boostability.
> >
> > Concerning the empirical results, unless I'm missing something, both Algorithms 1 and 3 are straightforward to implement. As such it's puzzling why no empirical result is provided in the paper, at least as far as error rates are concerned. I do agree though that runtime optimization is not in the scope of the paper.
> >
> > Also, the paper is still lacking a Conclusion/Discussion Section.

---

> > > ### Author Response · Authors · 2023-08-16
> > >
> > > Thank you for your response.
> > >
> > > The reviewer has requested to cast the WL conditions of previous works, e.g. [16] in the present framework similarly to how [16] cast Adaboost.MR in their framework.
> > >
> > > Notice that this comparison is not well defined.
> > > The reason is that in all these previous works there are strong assumptions over the data. For example, all past works that we are familiar with (e.g., [6], [16], Adaboost.MR etc.) assume there is a *known* hypothesis class that the data is realizable with respect to.  In this work we entirely remove that assumption.
> > >
> > > To clarify, is your question concerning the case that if we are forced to assume realizability w.r.t. a known class, then how do these two frameworks compare?
> > >
> > > We emphasize that our goal was to provide a framework that generalizes the binary setting, removing restrictive assumptions about the data while relying only on weak learnability. Our result is the first multiclass boosting framework  without these assumptions, building on binary boosting theory.

---

### Official Review · Reviewer_Rb54 · 2023-07-07

**Soundness:** 4 excellent
**Presentation:** 3 good
**Contribution:** 4 excellent
**Rating:** 8
**Confidence:** 3

**Summary:**

Boosting algorithms is an important fundamental question in SLT starting with celebrated AdaBoost algorithm. While the original results consider binary setting and the condition on the weak learner is straightforward, it is not trivial to formulate the same result for classification with more than one classes.

I did not have the chance to go through the proofs carefully due to time constraint. I do not see where there could be a potential mistake that cannot be fixed. I did not read through section 4 and I am not familiar with list learning.

pros:
- interesting fundamental problem
- the paper is well written
- a new condition for multiclass boosting
- relatively simple proof
- connection to list learning

cons:
- the only disadvantage I can see is that this paper would better fit other venues


minor comments and questions:
- should Theorem 1 be stated only for realizable case, since you relying on the compression scheme generalization bound?
- it is better to refer to Theorem 30.2 (than to Corollary 30.3) in Shalev-Swartz and Ben-David since your "outer" risk is equal to zero only with probability close to one
- perhaps this is obvious or I have missed it, but how does your condition compare to conditions of existing results for the multi-class boosting?

**Strengths:**

-

**Weaknesses:**

-

**Questions:**

-

**Limitations:**

-

---

> ### Author Rebuttal · Authors · 2023-08-10
>
> We thank the reviewer for their feedback and suggestions.
>
> - “Theorem 1 ” Notice that in our framework (in contrast to previous works on multiclass boosting) realizability assumption is not needed, and the empirical BRG assumption suffices to get the result. The reason is that by using compression bounds we only need to show that the boosting algorithm’s output is consistent with the training data, which is shown in the proof of Theorem 2.
> - “Theorem 30.2” we agree with the reviewer that this would be a better reference. We remark that in our current version of the paper we use a more accurate (and more general) statement of a compression bound that we prove in the paper. We will incorporate this in the final version of the paper as well.
> - “compare to conditions of existing results' ' - first we note that conditions of previous works, in particular [6,16] assume the PAC setting and realizability w.r.t a hypothesis class, whereas here these restrictions are removed. Moreover, these prior works require the WL to be agnostic and learn for any re-labeling of the data. Thus, these methods operate in the rather narrow setting of assuming realizability of the data, while also having access to an agnostic learner. However, it may be interesting to examine, when restricting to the PAC setting, how these conditions are related.

---

> > ### Comment · Reviewer_Rb54 · 2023-08-11
> >
> > Thank you for your answer.
> >
> > I am trying to understand why is it possible to have Theorem 2 for non-deterministic label. It seems the empirical BRG condition is not concerned with the underlying distribution at all, but the statement "for any $\mathcal{D}$ over $(X \times Y)$" in Theorem 2 does not add up.
> >
> > If we can guarantee $\Pr[H(x) \neq y] \leq \varepsilon$ for any $\varepsilon$, then we could not possibly have a noisy label. The risk value should be at least $1 - E \max_y \Pr(y | x) $.

---

> > > ### Author Response · Authors · 2023-08-12
> > >
> > > This is similar to binary boosting, where one can drop any explicit assumptions about the data, and instead only assume weak learnability holds (see discussion in Section 2.3.3 in the book "Boosting: Foundations and Algorithms" [19]).
> > >
> > > In essence, the weak learnability assumption implicitly asserts that the labels are not entirely noisy, and is in some sense a relaxed notion of the realizability assumption. For example, it removes the assumption that there is a *known* hypothesis class that generates the data. A similar idea is used here, where it suffices to only assume the BRG condition holds.
> > >
> > > Lastly, we note that indeed, the empirical BRG condition itself is not concerned with the underlying distribution. However, it is assumed in our theorem to hold for samples drawn from the underlying distribution.

---

> > > > ### Comment · Reviewer_Rb54 · 2023-08-12
> > > >
> > > > I see, thank you for clarification. I have no further questions.

---

### Official Review · Reviewer_bGbo · 2023-07-27

**Soundness:** 4 excellent
**Presentation:** 4 excellent
**Contribution:** 3 good
**Rating:** 7
**Confidence:** 3

**Summary:**

The paper proposes a novel treatment of traditional (non-gradient) boosting to multiclass prediction problems. In particular it shows that a novel relaxation of the weak learning criterion allows the definition of boosting algorithm with the usual success guarantee of reducing the empirical misclassification rate on the given data sample arbitrarily with high probability. This relaxation requires the base learner to also accept as “hint” a subset of labels and to, for each possible subset size k, to return a hypothesis that achieves an error rate better than 1-1/k with high probability. The proposed boosting algorithm makes use of this property by recursively eliminating candidate labels from the hints of each training example based on the hypotheses produced by previous iterations. Moreover, the paper shows a connection of the proposed theory to list-PAC learning and in particular gives a new characterisation of list-PAC learnability.

**Strengths:**

- The paper develops a beautifully simple extension of boosting to the multiclass case that, in contrast to previous approaches, does actually work to define a successful strong learner
- The connection to list learnability are very interesting
- The paper is well written and accessible

**Weaknesses:**

- Papers on classical boosting, in contrast to the additive ensembles produced by gradient boosting, feel rather narrow at this point. In particular, generalisations from misclassification rate to proper statistical loss functions are unclear.
- There is no empirical performance investigation of the proposed recursive boosting algorithm

**Questions:**

- In footnote 3 the authors state the assumption that m0 does only depend poly-logarithmically on m. It is unclear to me how strong an assumption that is and in fact also how sensitive the feasibility of the method is to violations of that assumption.
- My other questions is how to address the potential limitation listed below.

**Limitations:**

One potential limitation might be the definition of weak learners that satisfy the weak learning assumption. This is because it seems to require the restriction to different subsets of classes for different inputs. It is not immediately obvious to me how one would incorporate this, e.g., with a decision tree (if it depends on the concrete model at all).

---

> ### Author Rebuttal · Authors · 2023-08-10
>
> We thank the reviewer for their feedback and suggestions.
>
> - “Papers on classical boosting … narrow” - Gradient boosting methods are valuable algorithmic tools but lack a theoretical framework for studying the task of interest: multiclass classification, with respect to classification loss. Unlike binary boosting theory, gradient boosting methods cannot characterize sample complexity needed to reach a desired level of accuracy or reveal fundamental limitations. In contrast, our approach directly generalizes the theoretical boosting framework for multiclass classification that can answer these questions for general multiclass learning settings.
> Moreover, it does so through the classification loss itself, rather than proxy losses, as is the case for the gradient-based approach.
> - “empirical performance” - We note that, as the reviewer points out, our generalization provides a “beautifully simple extension of boosting to the multiclass case”. As such, and as discussed above,  we view the main contribution of this work to be the generalization of boosting theory. The algorithmic contribution is not the main focus of the paper. We leave the question of obtaining an optimal running time and performing empirical evaluations as an interesting future work. We certainly hope this will be further explored in a following line of work.
> - Footnote 3 - In the context of PAC learning, m_0 is only dependent on gamma, the edge of the WL. The comment says that the results hold even if m_0 is sublinear in m. We now see that this footnote is confusing and we will re-phrase it in the final version.
> - “weak learners that satisfy the weak learning assumption” -  observe that when given a list of, let’s say the top-5 choices of labels y per example x, we can “re-name” the labels of each example to be each position in the list! Therefore, we have effectively reduced the original label space to contain only 5 labels. That is, because the label names themselves don’t matter, we can do this re-naming and apply any standard learning algorithm as a WL (e.g., decision trees). We hope this answers the question, and will clarify this point in the paper as well.

---

> > ### Comment · Reviewer_bGbo · 2023-08-16
> >
> > Thank you for addressing my questions. My evaluation of the paper remains positive, but I still was not able to understand the practical construction of a weak learner that satisfies the required condition. I encourage the authors to clarify this even more in the final version.
> >
> > Further, as a complete side note, I do not fully agree with the characterisation of gradient boosting given in the rebuttal. In particular, note that the loss functions used in gradient boosting are not surrogate losses but correspond to log likelihoods of probabilistic response models. For classification, this is in a way more meaningful than just considering the zero/one loss because the loss values take the uncertainty of the Bernoulli response into account, which is fundamental if the model is supposed to be used for any practical purpose. Moreover, depending on the applied restrictions to the weight space, uniform convergence results should hold similar to those for 0/1 loss. Again all of this is a side note. I mentioned this "weakness" mainly to indicate that there would potentially be an even bigger interest for future results for gradient boosting. It takes nothing from the merit of the given excellent work.

---

> > > ### Author Response · Authors · 2023-08-22
> > >
> > > Thank you for your response. We agree that log likelihoods and related loss functions are important to study, as they provide valuable additional insights useful in practice. We also agree it would be an interesting future work to extend our current results to apply more broadly as the reviewer suggested.

---

### Official Review · Reviewer_Zpmo · 2023-07-28

**Soundness:** 3 good
**Presentation:** 2 fair
**Contribution:** 3 good
**Rating:** 4
**Confidence:** 1

**Summary:**

This paper proposes a generalization of boosting to the multiclass setting. To this end, the authors introduce a new weak learning assumption for multiclass based on a ‘hint’, which takes the form of a list of $k$ labels, named ‘Better-than-Random Guess’ (BRG). the authors present a main boosting method and provide theoretical analysis for PAC guarantees. Finally, the authors demonstrate applications based on the framework of List PAC learning.

**Strengths:**

- This paper proposes a new weak learning assumption, which encompasses both the binary case (i.e., $k=2$) as well as the cases with $k>2$.
- The main method is well-written.
- This paper provides abundant theoretical analyses.

**Weaknesses:**

- Although this paper is based on theories, it lacks experiments, including synthetic ones, and there is no conclusion.
- Due to the lack of experiments, there is insufficient comparison with previous works, with only half a page dedicated to it in section 1.2.

**Questions:**

Could you provide some experiments, including synthetic ones? It would be very helpful to understand this paper.

**Limitations:**

.

---

> ### Author Rebuttal · Authors · 2023-08-10
>
> We thank the reviewer for their feedback and suggestions.
> - “lack of experiments” - we emphasize that the main contribution of this work is the theoretical framework extending classical boosting theory to the multiclass setting. The algorithmic contribution is not the main focus of the paper. The question of finding the simplest and minimal assumptions on weak learnability as in the binary case has been studied in various works over the years, and in this work we provide the first intuitive and simple condition that leads to a simple boosting procedure. We leave the question of obtaining an optimal running time and performing empirical evaluations as an interesting future work. We certainly hope this will be further explored in a following line of work.
> - “comparison with previous work” - we refer the reviewer to the introduction, where we compare the current paper to various lines of work in the context of multiclass boosting. Moreover, the related work section covers other works in the broader context of boosting, starting from early extensions from the binary case and up to the most recent work in the area. The most related works, and most recent results on multiclass boosting theory, are [16] and [6], which are discussed in both sections.

---

### Decision · Program_Chairs · 2023-09-21

**Decision:**

Accept (poster)

**Comment:**

The work provides an original and rigorous analysis of Multiclass Boosting, notably by introducing a clever "better than random guessing" assumption based on a "hint" given to the weak learner. This kind of analysis is welcomed as it goes beyond the straightforward extension of existing works and thus fosters new ideas.

Nevertheless, I agree with the reviewers that it is unfortunate that the authors disregard empirical experiments. Although not required to justify acceptance, empirically evaluating the proposed "Recursive Boosting" method would have been natural. It might have aroused the interest of a larger audience, which is suited to a broad venue such as NeurIPS.

Finally, as I noted a few grammatical typos looking at the paper, I enjoin the authors to proofread their manuscript carefully. The References section should also be revised, as it contains a few arXiv bibliographic entries accepted in peer-reviewed venues.